# X-ray linear dichroic tomography of crystallographic and topological defects

Andreas Apseros[1,2 ✉], Valerio Scagnoli[1,2 ✉], Mirko Holler[3], Manuel Guizar-Sicairos[3,4], Zirui Gao[3,5,6], Christian Appel[3], Laura J. Heyderman[1,2], Claire Donnelly[7,8 ✉] & Johannes Ihli[3,9]

The functionality of materials is determined by their composition[1–4] and microstructure, that is, the distribution and orientation of crystalline grains, grain boundaries and the defects within them[5,6]. Until now, characterization techniques that map the distribution of grains, their orientation and the presence of defects have been limited to surface investigations, to spatial resolutions of a few hundred nanometres or to systems of thickness around 100 nm, thus requiring destructive sample preparation for measurements and preventing the study of system-representative volumes or the investigation of materials under operational conditions[7–15]. Here we present X-ray linear dichroic orientation tomography (XL-DOT), a quantitative, non-invasive technique that allows for an intragranular and intergranular characterization of extended polycrystalline and non-crystalline[16] materials in three dimensions. We present the detailed characterization of a polycrystalline sample of vanadium pentoxide ($V_2O_5$), a key catalyst in the production of sulfuric acid[17]. We determine the nanoscale composition, microstructure and crystal orientation throughout the polycrystalline sample with 73 nm spatial resolution. We identify and characterize grains, as well as twist, tilt and twin grain boundaries. We further observe the creation and annihilation of topological defects promoted by the presence of volume crystallographic defects. The non-destructive and spectroscopic nature of our method opens the door to operando combined chemical and microstructural investigations[11,18] of functional materials, including energy, mechanical and quantum materials.

Materials properties and functionality depend on the material composition and microstructure. The application-driven manufacturing of materials therefore requires an understanding of the underlying structure and composition, often at the nanoscale, as well as their link to the functionality. This is of great importance across many fields, including catalysis[3], energy storage and conversion[19,20] and permanent magnets[21]. Although compositional mapping down to the nanoscale, even across extended volumes, can be realized, for example, with high-spatial-resolution mapping of the electron density[11], characterization of the distribution, type and topology of crystallographic defects, and their dynamic behaviour under external stimuli, such as temperature, pressure or electromagnetic fields, remains a substantial challenge. To map the microstructure, including (crystal) grains and associated defects, and determine their structure–functionality relationship, 3D nanoscale mapping of the orientation within extended systems is key.

So far, high-spatial-resolution microstructure mapping has been possible with electron-based techniques such as transmission electron microscopy[7,8] and electron backscatter diffraction[13], achieving sub-10 nm spatial resolution with planar measurements. However, as these measurements are limited to materials of thickness on the order of 100 nm, destructive sectioning methods are required at present to acquire full 3D orientation maps of extended volumes. The non-destructive imaging of the crystallographic orientation of micrometre-thick materials has been addressed with diffraction contrast tomography[15] and dark-field microscopy[22]. However, these techniques are limited by the X-ray optics, with a typical spatial resolution of hundreds of nanometres, and so far have only been used to characterize highly crystalline samples. Crystal orientation and strain mapping has also been demonstrated with Bragg coherent diffractive imaging[23] and Bragg ptychography[10,24,25], by which it is possible to obtain a precise mapping of the crystal orientation with a spatial resolution of tens of nanometres, albeit in an effectively single-crystal object of low defect density[10,23–25].

Here we present the 3D mapping of nanoscale grains, grain boundaries and topological defects in polycrystalline $V_2O_5$ enabled by XL-DOT. By acquiring high-spatial-resolution synchrotron X-ray ptychographic projections of the linear dichroism, which is sensitive to the local crystal

[1]Laboratory for Mesoscopic Systems, Department of Materials, ETH Zürich, Zürich, Switzerland. [2]PSI Center for Neutron and Muon Sciences, Villigen, Switzerland. [3]PSI Center for Photon Science, Villigen, Switzerland. [4]École Polytechnique Fédérale de Lausanne (EPFL), Lausanne, Switzerland. [5]Brookhaven National Laboratory, Upton, NY, USA. [6]Department of Information Technology and Electrical Engineering, ETH Zürich, Zürich, Switzerland. [7]Max Planck Institute for Chemical Physics of Solids, Dresden, Germany. [8]International Institute for Sustainability with Knotted Chiral Meta Matter (WPI-SKCM2), Hiroshima University, Hiroshima, Japan. [9]University of Oxford, Oxford, UK. ✉e-mail: andreas.apseros@psi.ch; valerio.scagnoli@psi.ch; claire.donnelly@cpfs.mpg.de

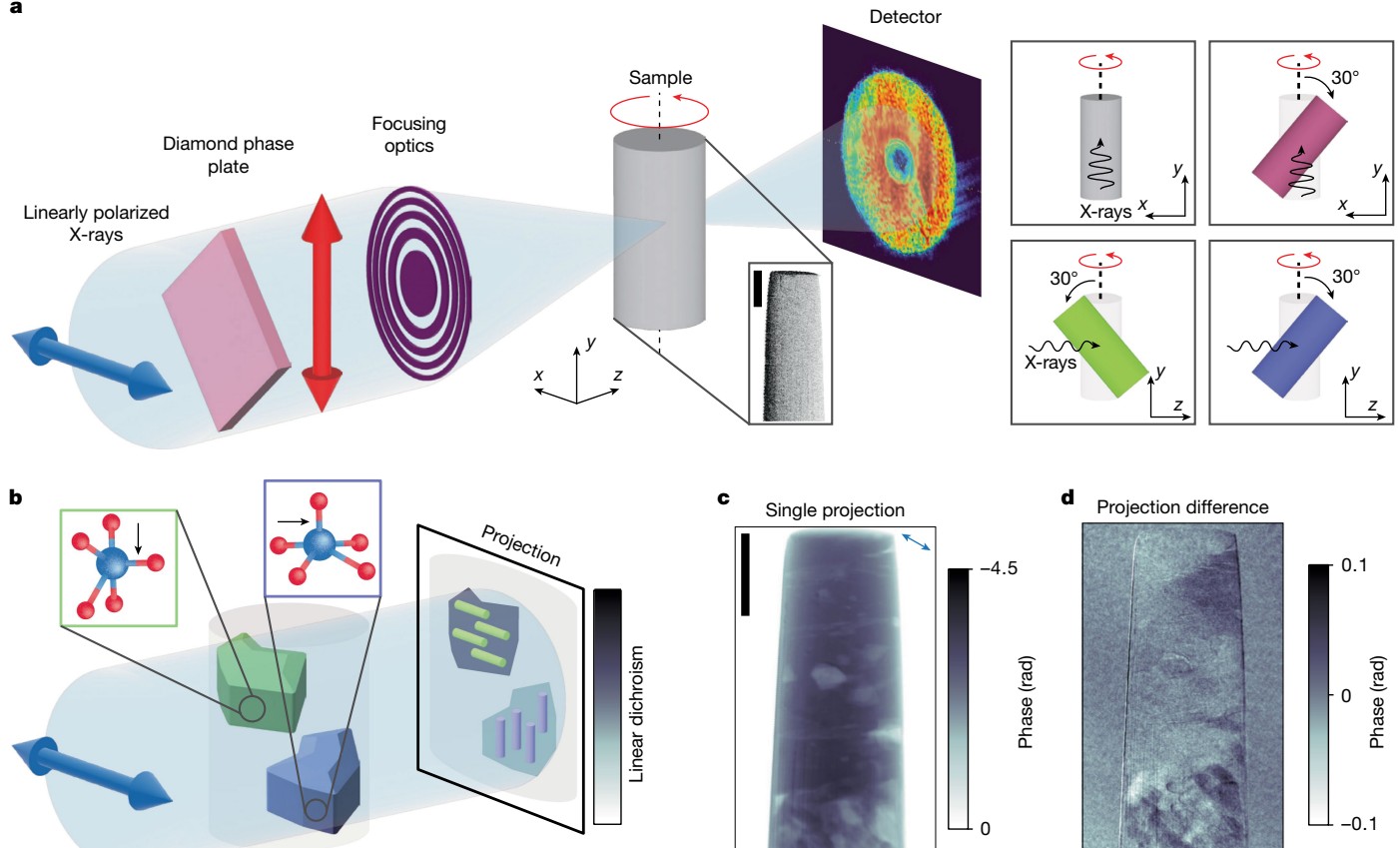

**Fig. 1 | XL-DOT experimental set-up. a,** Illustration of the XL-DOT set-up. Tomographic projections using LH (↔) and LV (↕) polarized X-rays were acquired through synchrotron X-ray ptychography. The sample was illuminated with focused X-rays using a Fresnel zone plate with engineered aberrations[50]. The sample was scanned in the *x*–*y* plane, with mutually overlapping illuminations. At each scanning point, a far-field diffraction pattern was acquired. A ptychographic reconstruction of these patterns returns phase and amplitude projections. A scanning electron micrograph of the studied cylindrical polycrystalline $V_2O_5$ sample is shown in the inset. A diamond phase plate was used to change the polarization of the X-rays. To examine and later reconstruct all three components of the crystallographic *c*-axis orientation, tomographic data were collected at four sample tilts, represented by the coloured cylinders in the four panels to the right, each with

LH and LV polarized illumination. The reference orientation is shown as a half-shaded grey pillar in all panels. **b,** Illustration of X-ray linear dichroism, showcased with the imaging of two grains of $V_2O_5$ (in green and blue). The transmission function of $V_2O_5$ under resonance conditions depends on the relative orientation of the Vanadyl bond (indicated with an arrow), which is parallel to the crystallographic *c* axis, and the linear polarization state of the illumination. Overlaid on the example projection are coloured rods that represent the orientation of the crystallographic *c* axis in the grains of the same colour. **c,** Phase-contrast projection of the sample acquired with LH polarization, whose relative orientation is shown by the double-headed arrow. **d,** the X-ray linear dichroic difference between projections acquired with LH and LV polarized illumination, illustrating electronic and linear dichroic phase-contrast contributions, respectively. Scale bars, 4 μm.

orientation, at different sample orientations and implementing a tailored reconstruction algorithm, we recover 3D composition and orientation maps of the examined material with nanoscale spatial resolution. The resulting tomogram provides the local orientation of the *c* axis of $V_2O_5$ (see Methods and Supplementary Note 3), allowing for the nanoscale segmentation of grains within the sample that are otherwise indistinguishable in the composition tomogram. This quantitative, non-invasive and simultaneous intragranular and intergranular characterization of extended polycrystalline and non-crystalline materials in 3D on the nanoscale is a key step towards the non-destructive operando imaging of material microstructure.

We demonstrate the capabilities of XL-DOT by mapping the underlying microstructure of a nanoporous $V_2O_5$, pillar prepared with high-temperature sintering, which is 6 μm in diameter and consists of randomly oriented, oxygen vacancy defect-rich, orthorhombic α-$V_2O_5$ crystals with an average grain size of more than 300 nm (Supplementary Figs. 1–3). The effectiveness of $V_2O_5$ as a heterogeneous catalyst in the chemical industry is closely related to its microstructure[17]. Therefore, a detailed mapping of the grain structure on the nanoscale will provide a route to understanding and optimizing its properties.

## Experimental methodology

To obtain a map of the grain orientation and defects in the polycrystalline sample, we determine the local orientation of the $V_2O_5$ crystallographic *c* axis in 3D by performing XL-DOT. The experimental set-up is shown schematically in Fig. 1a, along with an illustration explaining linear dichroic contrast in Fig. 1b. As opposed to conventional tomography, in which the acquisition of projections around a single tomographic rotation axis is sufficient to describe a scalar component, orientation tomography—similar to vector tomography[26]—requires the acquisition of projections around several tomographic rotation axes using a component-sensitive probe to access and characterize each of the three components of the orientation[26] (see Methods). Here, to gain sensitivity to the local *c*-axis orientation of the unit cell, we make use of X-ray linear dichroism, in which the transmission function of the sample depends on the relative angle between the incident X-ray beam polarization and the crystallographic *c* axis. High-spatial-resolution projections were acquired by investigating the linear dichroic component, and thus the local microstructure of the $V_2O_5$, with resonant dichroic X-ray ptychography[12,27,28]. For this, the energy of the X-rays

was tuned to the pre-edge of the V K-edge (5.468 keV). Using ptychography, phase projections were obtained, enabling the detection of the relatively weak linear dichroism signal[29,30] (Supplementary Figs. 4–6), as shown in Fig. 1c,d. Tomograms, consisting of 280 phase projections measured at regular angular intervals over 180°, were obtained for four sample tilts (see the four panels to the right of Fig. 1a), with both linear horizontal (LH) and linear vertical (LV) polarizations, resulting in a total of 2,240 phase projections. The tomographic projections were then aligned with high precision[31] and the 3D $c$-axis orientation tomogram—or map—was reconstructed using a gradient-based iterative reconstruction algorithm as described in the Methods and shown schematically in Supplementary Fig. 7. In the resulting $c$-axis orientation tomogram, we can resolve features of 40 nm in width, with a calculated average spatial resolution of 73 nm throughout the sample (see Methods and Supplementary Fig. 8).

As well as the $c$-axis orientation tomogram acquired on resonance, a high-spatial-resolution tomogram of the electron density was obtained with the X-rays tuned to an energy far from the absorption edge of 5.4 keV. There are no dichroic contributions to this non-resonant measurement, which provides a quantitative electron-density tomogram with a spatial resolution of 44 nm, allowing for local compositional analysis[32] (Fig. 2a–c and Supplementary Figs. 9 and 10).

## Compositional analysis

We first examine the off-resonance tomogram that provides a quantitative, high-spatial-resolution mapping of the sample's electron density, allowing us to identify the presence of different materials within the sample. The electron-density tomogram and a virtual cut are plotted in Fig. 2a,b, respectively. We find that, as well as solid regions corresponding to the electron density of $V_2O_5$, both nanoporous regions containing air and polystyrene, a residue of the synthesis process, are present, as identified quantitatively in the histogram in Fig. 2c. In the $V_2O_5$ regions, we observe two main forms. In the upper half of the sample (Fig. 2a), a large, continuous volume of $V_2O_5$ is observed. In the lower half of the sample, we can identify nanosized features whose orthorhombic shape reflects the favoured crystal phase of $V_2O_5$. Although the morphology seen in the electron-density tomogram indicates the presence of grain-like features such as the elongated grain highlighted in Fig. 2b, without knowledge of the crystallographic orientation, it is not possible to determine whether these are indeed single-crystal grains or volumes of polycrystalline material.

## Orientation tomography

The single-crystal and polycrystalline volumes can be distinguished by considering the 3D $c$-axis orientation map obtained with XL-DOT. Plotting the $c$-axis orientation map, shown in Fig. 2d,e, we immediately see a rich, complex structure with a large variety of local crystal orientations. First, we can confirm that the elongated grain identified in the electron-density map in Fig. 2b is indeed a single crystallite within our resolution (dashed black box in Fig. 2e, shown in full in Fig. 2j). Another example of a single crystallite grain is shown in Fig. 2g, and its location is marked with a dashed red box in Fig. 2e. However, we also observe regions that are polycrystalline, exhibiting abrupt changes in the crystal orientation that could not be easily distinguished in the electron-density tomogram. One example is marked by the solid red box in Fig. 2e and shown in full in Fig. 2f. Animated views of the morphology, as well as slices comparing the electron-density and orientation tomogram, are presented in Supplementary Videos 1 and 2, respectively.

The high-spatial-resolution map of the $c$-axis orientation allows us to detect grain boundaries associated with an angular mismatch down to 10° (Supplementary Figs. 11 and 12), a precision that, in principle, allows for the detection of more than 95% of grain boundaries in randomly oriented samples[33]. We use the orientation tomogram to segment grains

of distinct crystallographic orientations. Each separated volume represents a different $c$-axis orientation, as shown in Fig. 2h.

The successful segmentation of the crystallites, shown in Fig. 2h, also allows us to extract quantitative information about the morphology of the grains throughout the sample. The correlation between the size and shape of the grains is shown in the histogram in Fig. 2i. A strong negative correlation is observed between the size and shape of the particle, with larger grains exhibiting more elongated geometries. This is consistent with the orthorhombic morphology that is favoured by the growing crystal. With this segmentation, it is possible to observe that larger, more elongated grains (average sphericity = 0.26; example shown in Fig. 2j) up to 3.7 μm in length occur as free-standing objects in the more porous regions, whereas the smaller, more spherical grains (average sphericity = 0.3; example shown in Fig. 2k) of average length 1.3 μm are clustered in close-packed regions of the sample.

## Crystallographic defects

As well as the isolation of individual grains, the 3D $c$-axis orientation map allows for the identification of crystallographic defects, including grain boundaries and textures within regions that seem continuous in the electron density. For example, we can detect and differentiate between planar intergranular and intragranular crystallographic defects. In particular, it is possible to identify and map the local configuration of tilt, twist and twin grain boundaries within grains. We first consider a region in which the morphology seen in the electron-density tomogram (see Fig. 3a) indicates the meeting of two elongated grains. Within the limit of our spatial resolution, the electron density in the structure is homogeneous and no boundary can be observed. However, the $c$-axis orientation plot (Fig. 3b) reveals a distinct, abrupt grain boundary. The nature of this grain boundary becomes clear from the two grain orientations. In both grains, the $c$ axis is oriented at an angle of approximately 40° ± 6° to the grain boundary, exhibiting mirror symmetry across the boundary and thus indicating the presence of a twin defect schematically shown in Fig. 3c.

As well as investigating boundaries within free-standing grains, we can also take a closer look at the features that occur within the close-packed region in which the electron density seems homogeneous within the limit of our spatial resolution. Such a region is shown in Fig. 3d, in which—in the XL-DOT reconstruction image (Fig. 3e), also shown schematically in Fig. 3f—we can identify four regions with different $c$-axis orientations. In this polycrystalline region, two types of grain boundary are present. In the upper part, going from left to right across the boundary shown in Fig. 3g, the crystal orientation rotates from an in-plane ($x$–$y$ plane) to an out-of-plane orientation about a rotation axis parallel to the grain boundary normal: a prototypical twist grain boundary. By contrast, in the lower part, going from left to right across the boundary shown in Fig. 3h, the crystal orientation rotates abruptly from an in-plane ($x$–$y$ plane) to an out-of-plane orientation about a rotation axis perpendicular to the grain boundary normal: a tilt grain boundary. We note that these two examples of tilt and twist boundaries are associated with a high rotation angle of approximately 86° ± 6° and 65° ± 6°, respectively. A low-angle tilt grain boundary is also observed in the left-hand region (circled in Fig. 3e), facilitating the transformation of the grain boundary from twist to tilt.

## Topological defects

As well as grain orientation, XL-DOT reveals the presence of topological defects, which can subsequently be characterized. When considering the $x$–$y$ components of the orientation, we observe comet and trefoil defects[34] that also exist in nematic and biological systems[35]. These topological defects are associated with a local topological charge of +1/2 and −1/2 for the comet defect and trefoil defect, respectively (see Methods). In an infinite system, the overall topological charge must be

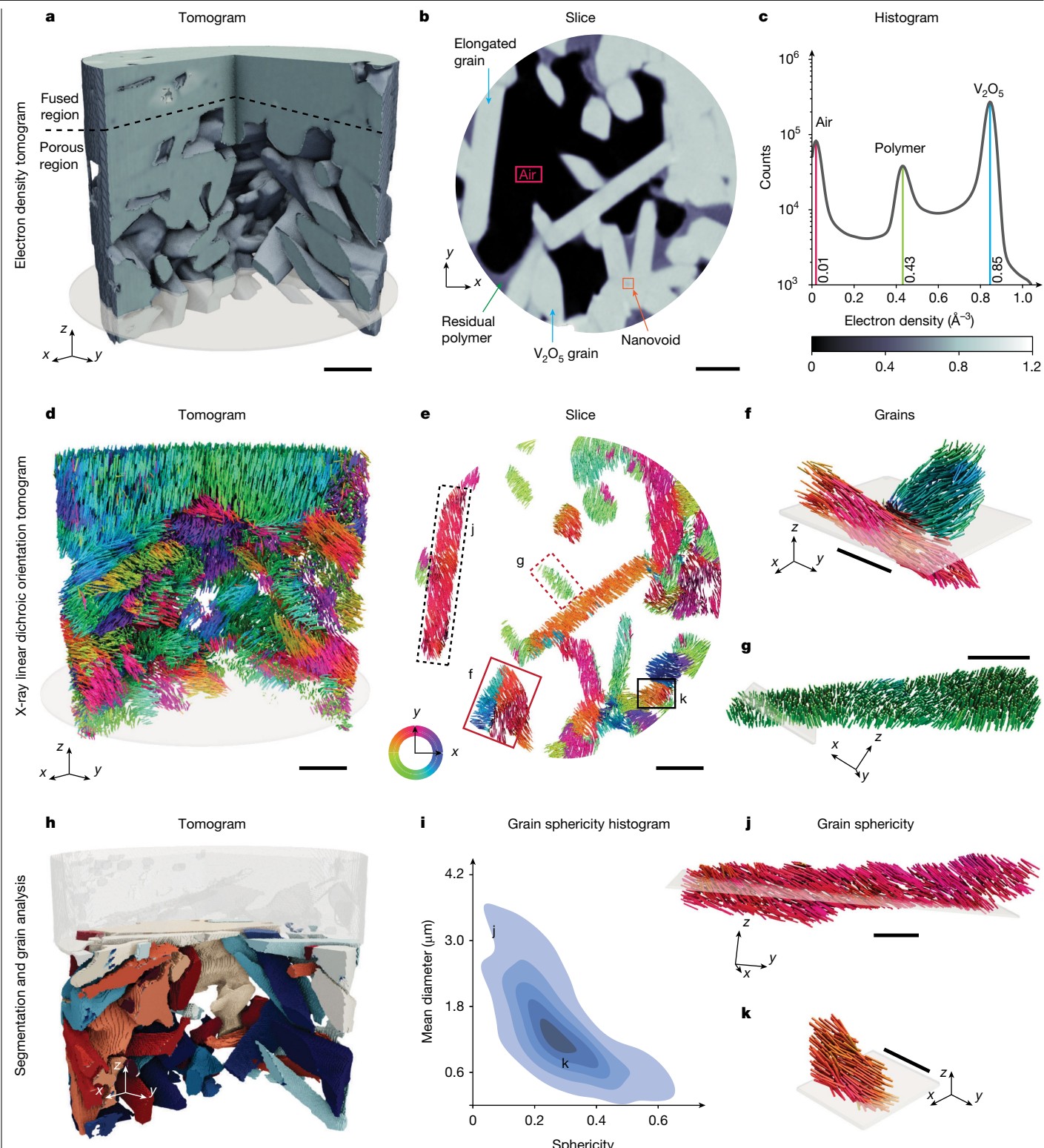

**Fig. 2 | 3D mapping the composition and microstructure of materials on the nanoscale with XL-DOT. a**–**c**, Electron-density tomography data. **a**,**b**, Volume rendering and horizontal slice through the tomogram reveal the constituent materials and morphology of the sample. **c**, Histogram of the electron density determined from the tomogram shown in **a**. The sample components, $V_2O_5$, polystyrene polymer and air, are indicated. **d**–**k**, XL-DOT reconstruction. **d**,**e**, Volume rendering and horizontal slice through the *c*-axis orientation tomogram, retrieved through XL-DOT, revealing the microstructure of the sample. Coloured rods represent the alignment of the crystallographic *c* axis in each voxel and are coloured according to their *x*–*y*-plane *c*-axis orientation. **f**,**g**, Volume rendering of selected regions of uniform electron density,

highlighting the ability to distinguish polycrystalline regions and single crystals with XL-DOT. **h**, Volume rendering of the *c*-axis orientation segmented tomogram, allowing for the identification of the sample's individual $V_2O_5$ grains. **i**, Quantitative analysis of the grain morphology reveals a strong correlation between the size and sphericity of the grains. **j**,**k**, Example of an elongated grain and a smaller grain with sphericity and mean diameter indicated in **i**. Further intergranular material characteristics, including electron-density correlations, are given in Supplementary Fig. 13. The locations of the highlighted grains (**f**, **g**, **j** and **k**) are indicated in **e** by the boxes. Transparent planes in **f**, **g**, **j** and **k** show the location at which the volumes intersect with the slice in **e**. Scale bars, 1 μm (**a**,**b**,**d**,**e**,**f**,**h**), 500 nm (**g**,**j**,**k**).

conserved and, thus, such defects can only be created or destroyed in pairs, or at surface boundaries.

In Fig. 4, we map the spatial evolution of crystalline topological defects in the vicinity of crystallographic volume defects. The layers shown in Fig. 4 are denoted a to e and colour-coded to match the corresponding boxes. In the lowest slice (a, red) a single trefoil defect (T1 in orange) exists in the vicinity of the large void. As we move upwards through the crystal (b, orange), we observe the trefoil defect T1 shift laterally such that its core is centred on a neighbouring nanovoid, with the surrounding topological structure maintained. Moving further upwards (c, green) to a layer above the nanovoid, the trefoil defect T1 is again laterally displaced into the crystal. In the next layer (d, blue), we observe the creation of a new trefoil defect (T2 in green) with a comet defect (C in blue) emerging from the surface of the large void. Because they are of opposite topological charge, the overall topology of the system is conserved. Notably, in the next slice (e, purple), the comet defect (in blue) and the original trefoil defect T1 annihilate, leaving behind a single trefoil defect T2. The animated spatial evolution of the trefoil and comet defects is provided in Supplementary Video 3.

The spatial creation and annihilation of topological defects in the $V_2O_5$ crystal is reminiscent of similar behaviour observed in the dynamics of topological defects in liquid crystals, biological systems and magnets. In particular, the recombination of the comet and trefoil defects is smooth and provides insight into the temporal evolution of the 3D crystal orientation during the high-temperature sintering process used to produce the sample.

The structural volume defects play an important role in the evolution of the topological defects. Because these are confined volume defects, they do not provide an infinite boundary, so that the overall topology of the system is conserved. Nevertheless, the interruption of the continuous crystal leads to a strong interaction between nanovoids and the trefoil defects. This can be seen with the pinning of the original trefoil (T1 in orange) at the smaller nanovoid (Fig. 4b). The core of the trefoil defect is associated with strong lattice bending or deformation, so its removal through the presence of the nanovoid minimizes the lattice energy. Moreover, the injection of the defect pair from the surface indicates that, although the overall topology must be conserved, the surface promotes the creation of defects in the system.

## Discussion and outlook

With the development of ptychographic XL-DOT, we demonstrate its application on a polycrystalline $V_2O_5$ pillar for which we not only map out the intragranular structure but also locate and identify nanoscale crystallographic and topological defects within the sample. As well as resolving twin, twist and tilt grain boundaries, we are able to map the presence and spatial evolution of comet and trefoil defects, revealing both the creation and the annihilation of defect pairs of opposite topological charge.

This capability to non-destructively map the composition and microstructure of extended polycrystalline samples in 3D on the nanoscale represents a substantial advance in the state of the art, in which—until now—such information was only available for isolated (single-crystal) samples of limited volume or through destructive or lower-spatial-resolution techniques.

XL-DOT takes an important position among present microstructural characterization techniques, bridging the gap between the high-spatial-resolution imaging of nanoscale samples with Bragg coherent diffractive imaging, Bragg ptychography, photoemission electron microscopy and electron-microscopy techniques and the lower-spatial-resolution imaging of larger samples accessible by diffraction contrast, dark-field tomography and tensor tomography[10,36,37]. With the ability to image larger samples at high spatial resolution, XL-DOT offers the non-destructive examination of nanoscale defects within system-representative sample volumes[7,10,12–15].

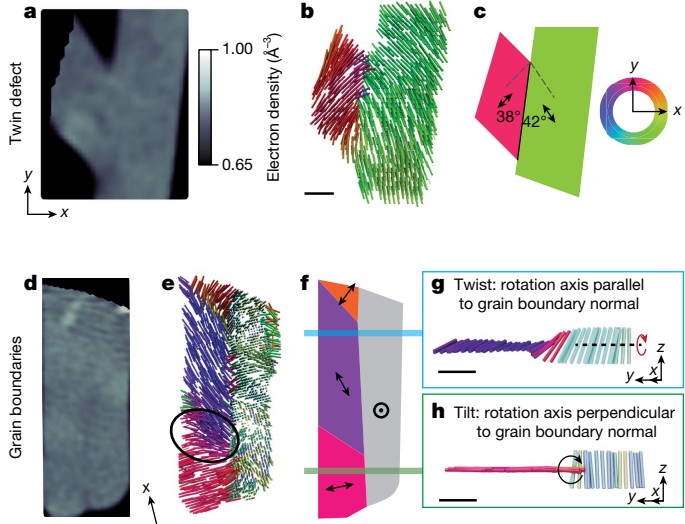

**Fig. 3 | Crystallographic defects detected using XL-DOT. a,d**, Electron-density renderings of selected regions showing a uniform electron density in the $V_2O_5$ sample. **b,e**, XL-DOT-extracted c-axis orientation of the same regions, revealing their complex internal microstructure. **c,f**, Schematic illustrations of the same region, indicating XL-DOT-detectable crystallographic defects in these areas. **a–c**, Twin grain boundary between two grains, with the c-axis orientation showing mirror symmetry about the abrupt grain boundary. **d–h**, Twist and tilt grain boundaries within a region of homogeneous electron density. XL-DOT reveals the presence of four grains with distinct c-axis orientations. Between the grains, we observe high-angle twist and tilt boundaries, also shown schematically in **g** and **h**, as well as a low-angle tilt boundary (circled in **e**). The orientation of the crystallographic c axis is indicated by the colour scheme, with the colour representing the x–y-plane orientation. The colours become lighter as the out-of-plane tilt increases. Electron-density renderings (**a,d**) share the same colour scale. Scale bars, 200 nm (**b,g,h**), 500 nm (**d**).

In contrast to the investigation of long-range order with diffraction techniques[8,10,13,15,22,25,37], XL-DOT examines the short-range order of a material through linear dichroism[16,29]. We note that, although the spectroscopic nature of XL-DOT presents further opportunities, the material of interest must exhibit X-ray linear dichroism and comply with X-ray transmission requirements (see Methods).

Linear dichroism is exhibited by a large variety of materials, ranging from non-cubic crystalline materials, to magnetic (ferromagnetic, ferrimagnetic and antiferromagnetic), ferroelectric and low-dimensional materials, to selected non-crystalline alloys, coordination complexes and molecular arrangements/networks[16,36,38–42]. Also, beyond mapping the orientation of non-cubic crystalline systems by exploiting the linear dichroism that they exhibit, the dichroic contrast mechanism can be used to map the orientation of the order parameter in non-crystalline materials (such as some ferromagnets) that—until now—could not be characterized with diffraction-based techniques alone[2,37,43]. This would, for example, allow—with nanoscale precision—the 3D characterization of (sectioned) biominerals[36,41], both the organic and inorganic parts, and synthetic polymers[16] in solar cells, as well as secondary phases located in grain boundaries or material inclusions within grains. For the biominerals and analogous systems, as investigations would involve tender and soft X-rays, for example, Ca and C K-edges, sample diameters are likely restricted to sub-5 μm in diameter. Although we focus here on the microstructural characterization of a material, the extension to linear magnetic dichroism is a natural progression[44,45] and will allow the characterization of previously inaccessible 3D magnetic topological textures in antiferromagnets[39].

A natural development of XL-DOT is the operando mapping of a material's composition and microstructure on the nanoscale. Here

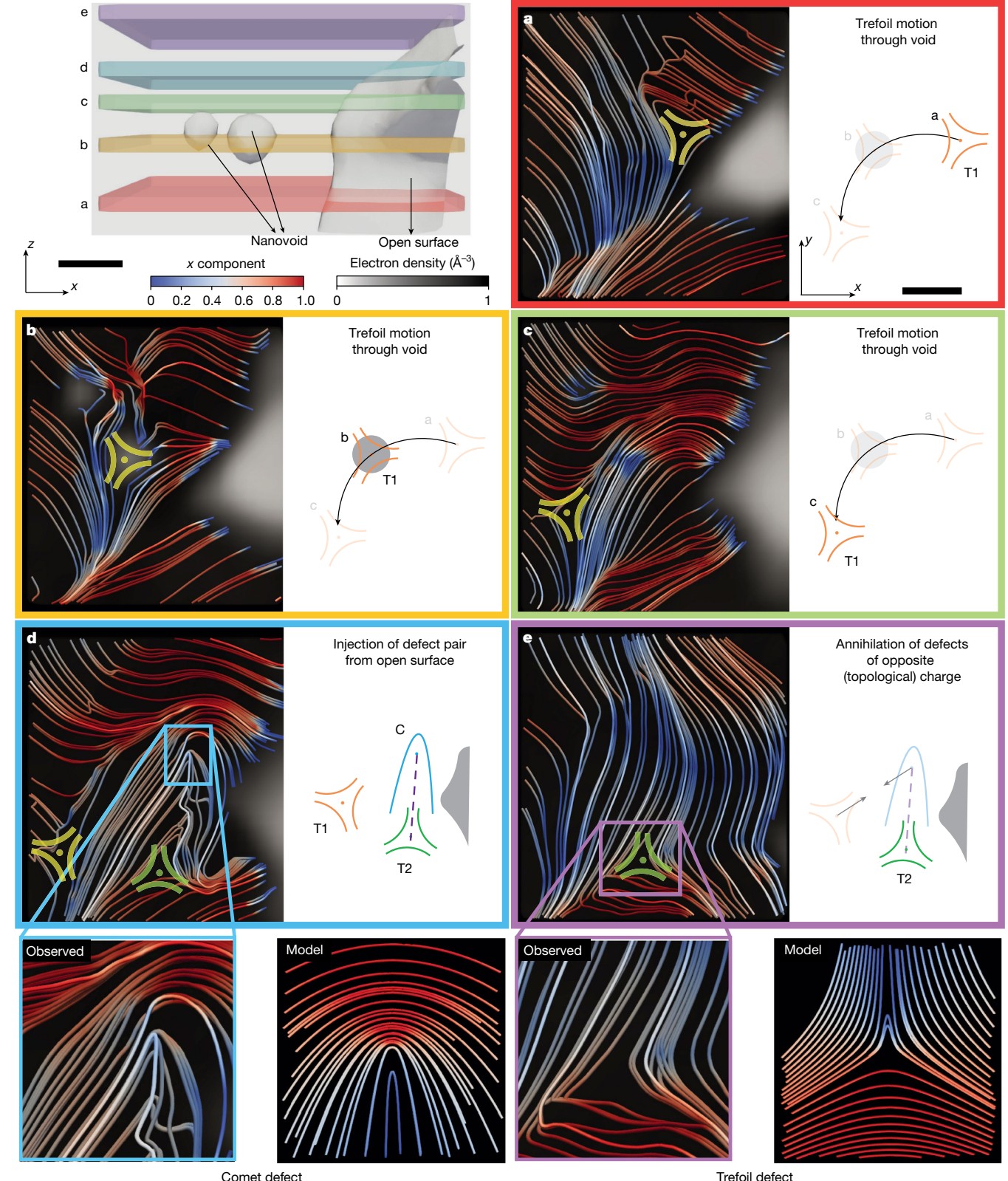

**Fig. 4 | Crystallographic volume and topological defects.** Top left, region of sample in which crystallographic volume defects and topological defects were observed, consisting of an open surface and two closed-surface nanovoids, indicated with white isosurfaces. Coloured slabs indicate the height of the layers shown by **a**–**e**. **a**–**e**, The evolution of the *c*-axis orientation through the thickness of the sample, with the creation and annihilation of topological defects. Streamlines represent the in-plane components of the orientation and are coloured according to their *x* component. They are overlaid on the electron density so that the location of volume defects is also visible (white regions).

**a**, A single trefoil defect (T1 in orange) is present near the open surface. Moving up through the thickness, the trefoil defect T1 shifts to be centred on a nanovoid (**b**), while maintaining its topology, and then it moves out of the nanovoid to the other side (**c**). **d**, A pair of dislocations with opposite topological charge (comet C in blue and trefoil T2 in green) are created at the open surface. **e**, The opposite charge pair consisting of the trefoil defect T1 and comet defect C annihilates and a single trefoil defect T2 remains. Bottom, higher magnifications of slices **d** and **e** showing experimental renderings of comet and trefoil topological defects (left) with model representations (right). Scale bars, 120 nm.

we can foresee following the evolution of the microstructure during material synthesis, under annealing conditions, albeit limited in temporal resolution by the XL-DOT acquisition time, on the order of a few hours. By performing quasi-static operando measurements, the exploration of microstructural changes under external stimuli, including the application of pressure and changes in atmosphere or temperature, will become possible. This ability to identify the role played by the microstructure on material performance will directly inform the improvement of future devices across a wide range of societally relevant industries. For operando measurements, the multi tilt-axis nature of this technique can prove challenging for complex sample environments. Nevertheless, we note that a preliminary reconstruction can be obtained with as few as two tilt axes (Supplementary Fig. 14). Moreover, we predict that the use of alternative geometries[46,47], which only require one tilt axis, will also allow for an easier implementation of operando imaging, as demonstrated for vector magnetic imaging[48].

Finally, with the use of coherent X-rays, XL-DOT stands to benefit notably from increases in coherent flux with the fourth generation of synchrotron radiation sources[49]. Not only will this greatly speed up measurements but it will also bring the spatial resolution down to the order of 10 nm, providing a way to map smaller point and line defects, such as dislocations. The increase in coherent flux will make even weaker dichroic signals detectable, making it possible to map the orientation of an even wider variety of materials in 3D.

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

## Methods

### Materials

We purchased $V_2O_5$ from US Research Nanomaterials and polystyrene latex spheres, 330 nm in diameter, from Thermo Fisher Scientific.

### Sample preparation

The examined pillar was extracted from a sintered millimetre-sized pellet. This was prepared from a mixture of nanocrystalline $V_2O_5$ and polystyrene spheres (85/15 wt%). Mortar and pestle (10 min) was used to homogenize the mixture before the pellet was pressed using a 17 mm die set (3 min, 1.2 t uniaxial load). To increase the $V_2O_5$ grain size, sinter and create the desired porous structure, we heated the pellet to 590 °C for 5 h (Supplementary Fig. 1 and Supplementary Note 2). The polycrystalline $V_2O_5$ pillar was prepared by mechanically fracturing the sintered pellet, after which a fracture piece was mounted on an OMNY tomography pin[51] using epoxy resin. The pillar was then pre-shaped using a microlathe[52], before being reduced in diameter to 6 μm using focused ion beam (FIB) milling. This pillar was then transferred onto a second OMNY pin[51]. The tip of the second OMNY pin was sharpened using FIB milling before transferring the pillar. This was necessary to facilitate tomography measurements with a 30° stage tilt[26]. See Supplementary Fig. 3 for micrographs of the prepared pillar.

### General material characterization

Scanning electron microscopy and FIB milling were performed using a Zeiss NVision 40 dual-beam FIB. Powder X-ray diffraction measurements of the sample before and after sintering were acquired using a Cu K-α radiation source with a step size of 0.02° $2\theta$, (refs. 53,54) (Supplementary Fig. 2). The sintered sample consists of α-$V_2O_5$ with a grain size >100 nm.

### Origin of linear dichroism in α-$V_2O_5$

$V_2O_5$ has a layered orthorhombic crystal structure consisting of distorted [$VO_5$] pyramids, shown schematically in Fig. 1b. These pyramids tile along the $a$−$b$ plane and are bound with van der Waals interactions along the $c$ axis. The apical, vanadyl bond of these pyramids, aligned with the crystallographic $c$ axis, is shorter (1.57 Å) compared with the bonds on the base of the pyramid (1.87 Å). This shorter bond breaks the symmetry of an otherwise regular square pyramid[55]. To examine the spatial orientation of the apical bond and, in turn, the orientation of entire grains and deviations within them, the energy of the incident X-rays was set to that of the vanadium K pre-edge peak[55]. This peak arises from the V (1 s) to V (4p-3d) transition, more specifically, to V (3d $e_g$ + $4_p$) + O $2p_z$ mixing states, which become accessible as a result of the deviation of the V coordination from the octahedral symmetry. When the apical bond is parallel to the direction of the electric field of the incident X-rays, the interaction is strong, as the transition V (1 s) to V (4p-3d) is allowed. When the apical bond is instead perpendicular to the incident polarization, the interaction is weaker[29,55]. An illustration of the different absorption strengths that result from the relative orientation between the incident X-ray polarization and the apical bond, known as linear dichroism, is shown in Fig. 1b. The X-ray near-edge absorption and phase spectra of $V_2O_5$ measured using LH and LV polarizations are shown in Supplementary Fig. 4 and a schematic of the layered crystal structure is provided in ref. 29.

In the above-described relationship between the polarization state of the illumination and the examined asymmetry or anisotropy, the linearly polarized light acts as a 'search light' for the resonant bond to which the polarization is parallel. This relationship applies, in principle, to all cases of natural linear dichroism[16,42]. The connection between the investigated anisotropy orientation and the unit-cell orientation of the material can be obtained through the use of reference samples, as showcased in 2D linear dichroic microscopy applications[36], and is already available in the literature for numerous materials. It can also be readily determined with previous knowledge of the material's crystal structure (or molecular arrangement)[55].

### Ptychography, PXCT and phase contrast

Ptychography is a lensless imaging technique in which the phase problem is solved by means of iterative phase-retrieval algorithms[27]. By applying ptychography to solve the phase problem at different projection angles, its tomographic extension, ptychographic X-ray computed tomography (PXCT)[56], is able to retrieve the complex-valued transmissivity of the specimen, providing quantitative tomograms of both phase and amplitude contrast[32]. Both the individual images−or projections−and resulting tomograms obtained using X-ray ptychography are sensitive to changes in the complex-valued refractive index, $\eta$. The real part of the refractive index decrement, $\delta$, corresponds to the phase, whereas the imaginary part of the refractive index corresponds to the amplitude, $\beta$. The refractive index is fundamentally an expression of the complex atomic scattering factor, $f = f_1 + if_2$. The refractive index is therefore given by:

$$n = 1 - \delta - i\beta = 1 - \frac{r_e}{2\pi}\lambda^2 \sum_k n_{at}^k (f_1^k + if_2^k), \tag{1}$$

with $r_e$ being the classical electron radius and $\lambda$ the illumination wavelength[57,58]. The images and tomograms resulting from measurements performed with incident X-ray energies away from sample-relevant absorption edges can, in the case of tomograms, be converted to quantitative electron-density, $n_e$, and absorption index, $\mu$, tomograms[32]. Measurements conducted near sample-relevant absorption edges, that is, examining specific electronic transitions and the associated increase in the photoabsorption cross-section, are subject to anomalous scattering effects[57,58], including dichroism.

The angular dependence of the linear dichroism has previously been used in a microscopy context, in particular in X-ray linear dichroism microscopy with secondary imaging modalities such as photoemission electron microscopy, to provide a 2D spatially resolved microstructural characterization tool[16,29,30,36,59,60]. The reader is directed to the initial work of Ade and Hsiao[16] and the more recent works of Gilbert et al.[36,61–63] and Collins et al.[59,60,64]. In the present work, we have developed the capability to map the orientation in 3D by combining X-ray linear dichroism microscopy with PXCT (XL-DOT).

Although XL-DOT can be applied with a range of imaging techniques, such as scanning transmission X-ray microscopy, we have selected X-ray ptychography as the imaging modality, a choice motivated by three factors. (1) PXCT provides quantitative or absolute contrast tomograms, which is ideal for material or component identification and for the detection of marginal signal variations[11,30]. (2) As a lensless imaging technique, ptychography excels in terms of signal-to-noise ratio (SNR), spatial resolution and dose efficiency (per resolution element) compared with other methods[65–68]. Given its superior SNR, it is ideal for measuring the relatively weak linear dichroism signal exhibited by $V_2O_5$ (refs. 30,61). (3) Ptychography can access the phase component. Phase changes at the vanadium K-edge are twice as large as changes in the absorption, so that the retrieved phase projections have a higher spatial resolution and superior SNR; see Supplementary Figs. 5 and 6 (ref. 58). We performed all data analysis on the phase component of the projections and tomograms only.

### Ptychographic linear dichroic X-ray tomography

**Data acquisition.** Experiments were carried out at the coherent small-angle X-ray scattering (cSAXS) beamline of the Swiss Light Source. The photon energy was selected using a double-crystal Si(111) monochromator. The horizontal aperture of slits located 22 m upstream of the sample was set to 20 μm, creating a virtual source point that coherently illuminates a 220-μm-diameter Fresnel zone plate with an outermost zone width of 60 nm and with engineered aberrations

designed to improve reconstruction contrast and spatial resolution[50]. Coherent diffraction patterns were acquired using an in-vacuum Eiger 1.5M area detector, with a 75 µm pixel size, placed 5.235 m downstream of the sample inside an evacuated flight tube. Tomography experiments were performed using the positioning instrument described in ref. 69.

To map the local orientation of the apical bond within the examined sample volume in 3D, we exploited its linear dichroism and acquired eight equiangular ptychographic tomograms over 180° at 5.469 keV for different illumination polarizations and sample tilts. Specifically, ptychographic tomograms were acquired with a LH and LV polarization of the incident illumination at 0° stage tilt and at 30° stage tilt (sample in grey and pink in the top two panels on the right of Fig. 1a). Two further tilts were measured, whereby the sample was first rotated by +90° and −90° about the main axis of the pillar, followed by a 30° stage tilt[26]. The last two tilts are equivalent to tilting towards and away from the beam by 30° (sample in green and blue in the bottom two panels on the right of Fig. 1a). Examination under different sample tilts and X-ray polarizations is required to have sufficient information for the construction of an orientation tomogram representative of the apical bond orientation in 3D[26,47]. To change the illumination source native horizontal polarization to vertical, we used a 250-µm-thick diamond crystal phase plate inserted into the illumination path upstream of the zone plate (see Fig. 1a). The phase plate absorbed approximately 65% of the incident photons[70]. The degree of polarization of the X-rays was determined to be approximately 60% using a polarization analyser set-up. The sample tilt was changed using a sample holder insert[26]. To minimize the acquisition time, we used an adaptive field of view for each group of ptychographic projections. The maximum field of view, horizontal × vertical, was about $24 \times 25$ µm$^2$. The scanning followed a Fermat's spiral pattern[71]. An average step size of 0.8 µm was used for all tomograms. The exposure time per scanning point was 0.1 s. 280 projections were acquired per tomogram.

Finally, using the same acquisition parameters, we acquired an off-resonance ptychographic tomogram of the pillar below the absorption edge at 5.4 keV. This tomogram, being insensitive to any dichroic effects, was used for computing the electron-density tomogram and subsequently used for compositional analysis[11]. It should be noted that the starting angle and angular spacing of projections was kept constant across all tomograms.

**Ptychographic image reconstruction.** Ptychographic images (or tomographic projections) were reconstructed using the PtychoShelves package[72]. For each reconstruction, a region of $600 \times 600$ pixels of the detector was used per scanning point, resulting in an image pixel size of 30.91 nm for the pre-edge and 31.29 nm for the below-edge tomogram. Reconstructions were obtained with 200 iterations of the difference map algorithm[73], followed by 300 iterations of maximum likelihood refinement[74].

**Preprocessing of projections.** Before any tomogram reconstructions, we: (1) resampled all projections to a pixel size of 30.91 nm using Fourier interpolation; (2) extracted the phase from the reconstructed projections, removed constant and linear phase components and spatially aligned the projections using a tomographic consistency approach[31]; and (3) aligned all projections to a common pillar orientation. As a last step, the different orientations at which projections were measured were characterized by a 3D rotation matrix[26], which was input into a specially developed reconstruction code (see the 'XL-DOT reconstruction' section below). It should be noted that, owing to the sample tilt and the fixed vertical field of view of the 2D projections, the 3D volume that is commonly sampled in all orientations, and used in the subsequent analysis and visualization, is reduced. (4) Last, to isolate the dichroic component from the isotropic electron-density contribution, the LV projection was subtracted from the LH projection. The resulting set of

projections were used in the reconstruction of the XL-DOT dataset, as discussed further below.

**Ptychographic tomogram reconstruction.** The ptychographic tomogram, acquired with the X-ray energy tuned to below the absorption edge, was reconstructed using a modified filtered back-projection algorithm[75]. This off-resonance phase tomogram was used to derive the electron-density tomogram, which was then used for material component identification[11,32].

**XL-DOT reconstruction.** A gradient-based iterative reconstruction algorithm was developed to reconstruct the orientation field in 3D. A schematic of the reconstruction process is shown in Supplementary Fig. 7. The process starts with the creation of a 3D starting, random guess of the sample. Using the sample–illumination interaction relationship in equation (2), a set of projections is simulated. These projections are then compared with the measured set of projections and their difference is used to compute a gradient to iteratively correct the initial guess.

The interaction between the electric field of the incident linearly polarized X-rays, $\vec{E}$, and the orientation of the apical vanadyl bond, $\vec{a}$, can be described as:

$$f = f_0 + f_{\text{lin}} (\vec{E} \cdot \vec{a})^2 \tag{2}$$

Here $f$ is the total scattering factor, which contains the isotropic charge contribution, $f_0$, and the linear dichroism contribution, $(\vec{E} \cdot \vec{a})^2$, with a pre-factor $f_{\text{lin}}$ that depends on the electronic transition under resonance. Keeping with the experimental geometry (Fig. 1a); using X-rays with a LH polarization parallel to the $x$ axis and denoting an arbitrary polarization angle as $\varphi$, in which $\varphi = 0°$ is LH polarization and $\varphi = 90°$ is LV polarization, the tomographic rotation and tilting of the sample can be quantitatively represented by the 3D rotation matrix **R**. In transmission, the measured projection can then be described by the integral given in equation (3). Index summation notation is used to give the rotation of the relevant components of the orientation, $a_j$. The integration is evaluated along the X-ray propagation direction, the $z$ axis.

$$P(x,y) = \int f_0 (\mathbf{R}\vec{r}) + f_{\text{lin}} [R_{1j} a_j (\mathbf{R}\vec{r})\cos\varphi + R_{2j} a_j (\mathbf{R}\vec{r})\sin\varphi]^2 \mathrm{d}z \tag{3}$$

Knowing the form of the interaction, the reconstruction algorithm was formulated by generating a guess structure, from which projections were simulated at the same orientations that the sample was measured. These simulated projections, $\hat{P}$, were then compared with the corresponding measured projections, $P$. Their square difference was used to define an error metric, $\epsilon$, quantifying how well the guess could reproduce the measured projections, given by

$$\epsilon = \sum_{m,x,y} [\hat{P}^m(x,y) - P^m(x,y)]^2 \tag{4}$$

in which $m$ represents the projection index. The error metric was reduced using gradient descent, therefore improving the ability of the guess structure to represent the internal $c$-axis orientation of the measured sample. By differentiating the error metric in equation (4) with respect to each component, we obtain the following analytical expression for calculating the gradient:

$$\frac{\partial \epsilon}{\partial a_k} = 4 f_{\text{lin}} \sum_{x,y} [\hat{P}^m(x,y) - P^m(x,y)]$$
$$[R_{1j} a_j \cos\varphi + R_{2j} a_j \sin\varphi](R_{1k}\cos\varphi + R_{2k}\sin\varphi) \tag{5}$$

The gradient was evaluated and applied to the guess structure at every iteration. During the reconstruction process, the magnitude of

the linear dichroic contrast, corresponding to $f_{lin}$, was not constrained and was therefore also optimized during gradient descent. As a result, it is not necessary to predetermine the $f_{lin}$ value. As the iterative gradient descent reconstruction is prone to converging at local minima, 40 individual reconstructions were performed using different random, non-zero initial conditions. The individual reconstructions are combined by averaging all components to obtain a final reconstruction. The difference in the angular orientations between the individual reconstructions and the final, averaged reconstruction was used to evaluate the standard deviation of the orientation, which is an estimate of the uncertainty in orientation.

Notably, using equation (3), it can be shown that LV polarization ($\varphi = 90°$) projection measurements evaluate to

$$P(x,y) = \int (f_0(\mathbf{R}\vec{r}) + f_{lin}[\mathbf{a_y}(\mathbf{R}\vec{r})]^2)dz \qquad (6)$$

Because there are no vector rotations in this expression, it is equivalent to examining a scalar consisting of two components: the isotropic charge background, $f_0$, and the (out-of-plane) $a_y^2$ component. This can be reconstructed with conventional tomography and gives contrast between grains that are in-plane ($x$–$y$ plane) and out-of-plane oriented. This contrast was used for further validation of the final reconstruction, as shown in Supplementary Fig. 12.

**Multiaxis tomography.** To obtain a first estimation of how many sample tilts and linear polarization states are necessary for a robust XL-DOT reconstruction, we performed a series of numerical simulations and tomographic reconstructions with fewer sample tilt axes (Supplementary Fig. 14). Preliminary reconstructions can be obtained with as little as two tilt axes using LV and LH polarizations only. Both our simulations (not shown) and recent literature[30,47] indicate further that the numerous tilt axes can be replaced by measurements with extra X-ray polarizations[76]. Similar results can also be achieved using laminography[46,48]. This offers a route to fewer or even single tilt-axis measurements.

**Dose estimation.** The total deposited dose over the duration of the experiment and the entire volume of the $V_2O_5$ pillar was approximately $10^9$ Gy. This estimate is based on the mass density of the sample and the average flux density per projection[77]. No actions were taken to limit the dose, as $V_2O_5$ is not known to degrade under the present experimental conditions[11,29]. For radiation-sensitive materials, preventative measures can be used to mitigate or account for potential radiation damage[78]. Dose-limiting options include scanning and projection sparse acquisition schemes[11,79] that reduce the total deposited dose, changes to the ptychography acquisition such as using an out-of-focus acquisition with micrometre-sized scanning probes which lead to a reduction of both the total and peak dose per area, as well as the implementation of cryogenic and inert atmosphere measurement conditions[80,81].

**Spatial resolution.** Spatial resolution estimates of projections and tomograms were obtained using Fourier ring correlation and Fourier shell correlation, respectively[82].

To evaluate the spatial resolution of the acquired projections, we acquired projections under identical conditions, that is, at the same rotation angle, calculated the correlation between these two images in the Fourier domain and estimated the spatial resolution based on the intersection with a one-bit threshold (see Supplementary Fig. 6). This gives spatial resolutions close to the pixel limit of 30.91 nm and 31.29 nm for the on-resonance (5.469 keV) and off-resonance (5.4 keV) measured projections, respectively.

To evaluate the spatial resolution of the electron-density tomogram acquired below the absorption edge, we halved the entire dataset and reconstructed two independent tomograms (Supplementary Fig. 10). This gives a 3D spatial resolution of 44 nm.

To evaluate the spatial resolution of the orientation vector field, the corresponding dataset was similarly split in half and two tomograms of the orientation vector field were calculated. Using Fourier shell correlation, we calculated spatial resolution estimates for each of the orientation scalar components ($LD_x$, $LD_y$, $LD_z$), as shown in Supplementary Fig. 8, providing a lower bound for their spatial resolutions of 84 nm, 45 nm and 89 nm, respectively. Also, we measured edge profiles across sharp features such as 90° grain boundaries, which revealed a maximum edge sharpness of 40 nm, with an average edge sharpness of 73 nm, which we take as the spatial resolution of the orientation tomogram.

**Measurement error estimation.** To estimate the voxel-level electron-density uncertainty, we calculated the standard deviation ($\sigma$) of the electron density in a region of air surrounding the imaged pillar. The average electron density in air and uncertainty was calculated as $0.004 \pm 0.007$ Å$^{-3}$.

To estimate the uncertainty in the detected linear dichroism, that is, spatial variations in the pre-edge peak intensity, we independently reconstructed the LV and LH phase tomograms with the sample at a fixed sample tilt and then subtracted them from each other. We then isolated a region of air and calculated the standard deviation in the phase shift associated with the voxels in this region. This standard deviation of the phase associated with the air region corresponds to the uncertainty of the dichroic signal. On the basis of this procedure, the uncertainty of the dichroic signal is found to be $1.3 \times 10^{-4}$ rad, which corresponds to a refractive index decrement, $\delta$, error of $1.9 \times 10^{-7}$.

To estimate the error in the determined orientation, we isolated an elongated grain with a volume of 0.85 μm$^3$ and long-edge length of 3.2 μm that showed the least variance in electron density and $V_2O_5$ orientation, that is, which is assumed to be single crystal, and calculated the standard deviation ($\sigma$) in orientation to be ±10° for azimuth ($x$–$y$-plane angles) and ±8° for elevation (out-of-plane angles) (Supplementary Fig. 11).

The critical concentration for element detection can be estimated to correspond to a dichroic magnitude (difference between tomograms taken with different polarizations) of at least twice the reconstruction error. The dichroic contrast of the $V_2O_5$ is $1.8 \times 10^{-3}$ and the noise in the reconstruction is an order of magnitude weaker at $1.3 \times 10^{-4}$. As a result, in $V_2O_5$, our dichroic contrast is 12 times the error. We can estimate that, if all other parameters are held constant, the concentration of V can be decreased by a factor of 6 and still be measurable.

**Present XL-DOT acquisition time and future prospects.** The total acquisition time for the XL-DOT dataset used in this work was around 85 h, including sample tilting, changing the polarization and alignment and dead-time overheads. The pure measurement time, however, was only about 24 h. This discrepancy is largely because of the lack of automation. There exist several opportunities to reduce the acquisition time as follows:

1. Reduce oversampling: reconstructions using 50% of the tomograms provide similar results (Supplementary Fig. 14).
2. Automation and imaging geometry: the measurement of intermediate linear X-ray polarization angles[30,47,70,76] and/or use of the laminography geometry[46,48] will eliminate most of the present acquisition overheads.
3. The increase in coherent flux expected from fourth-generation synchrotron light sources promises to reduce scan times for radiation-hard materials[83].
4. Further innovations such as multibeam ptychography and sparse tomography offer routes to even faster data acquisition[11,84], providing acquisition times compatible with operando measurements[48,85].

### Data analysis

Analysis of the dichroic tomogram was performed using in-house-developed MATLAB routines, ParaView and Avizo. To account for the

damage caused during the FIB milling step of the sample preparation, we defined a mask that excluded the outermost 90 nm of the sample cylinder from orientation and electron-density volume analysis (Supplementary Fig. 9).

**Component identification and isolation.** Materials were identified by comparing the tabulated electron densities of the known sample and reference components, listed in Supplementary Table 1, with the PXCT-measured electron densities. Shown in Supplementary Fig. 9 is a volume rendering and a horizontal cut slice through the electron-density tomogram with the corresponding electron-density histogram. The $V_2O_5$ volume was isolated using threshold segmentation with a lower bound of 0.74 $Å^{-3}$ and an upper bound of 0.90 $Å^{-3}$.

**Analysis of topological defects.** The topological charge can be determined by considering the winding number associated with a given topological defect. The winding number corresponds to how the crystallographic orientation changes when moving around a circle enclosing the defect in a clockwise manner. For the comet (trefoil) defect, the $c$ axis rotates clockwise (anticlockwise) by +180° (−180°) for one complete revolution. As the crystallographic orientation has completed half a revolution of a full circle (360°), the topological numbers ±1/2 are assigned to them.

**Microstructural analysis of $V_2O_5$ domains.** To isolate the $V_2O_5$ grains and facilitate a correlation between orientation and electron density, we applied the above-defined threshold mask (electron densities between 0.74 $Å^{-3}$ and 0.90 $Å^{-3}$) to the orientation tomogram. To identify and characterize individual $V_2O_5$ grains, we downsampled the masked XL-DOT reconstruction by a factor of three (transforming a group of 3 × 3 voxels into 1 voxel with an average intensity value of the same size), thus reducing the sensitivity to intragranular variations. Segmentation was then performed by separating regions along high-angle grain boundaries, showing a $c$-axis orientation difference larger than 10°. Following segmentation, we then calculated the volume of these grains, their mean diameter and their sphericity[86]. Shown in Supplementary Fig. 13 are the corresponding distributions and correlations of the segmented grains.

**Sample diameter and photon energy resolution considerations.** As linear dichroic phenomena occur near absorption edges or resonant X-ray energies, the X-ray penetration depth at these energies determines the sample diameter that can be investigated with XL-DOT. For most materials, it is the penetration depth at the X-ray energy of the examined chemical element that sets an upper limit on the sample diameter. Taking pure transition metals as an example, this imposes a typical upper limit to the sample size of around 10 µm. Transition-metal-rich functional materials such as catalyst bodies, cathode materials, ferroelectrics, biominerals and concrete, which are also of interest for XL-DOT measurements, exhibit a substantially larger upper sample size limit owing to their internal porosity or composite nature. For instance, a 100 µm-thick $V_2O_5$ sample transmits around 10% of the incident beam in the pre-edge region (https://henke.lbl.gov/). 3D or nanotomography measurements of such sample diameters are increasingly typical for operando measurements[48,87–89].

Although XL-DOT measurements should ideally be performed at the X-ray energy of an absorption edge at which linear dichroic contrast is strongest to maximize contrast in the projections, the range of energies at the absorption edge at which dichroism can be measured can be large. For instance, the full width at half maximum of the near-edge peak in our $V_2O_5$ spectra used for XL-DOT is approximately 3 eV, which means that even an X-ray energy resolution of up to 3 eV would be sufficient for XL-DOT measurements, albeit at a decreased SNR. There is therefore a degree of flexibility in terms of the required energy position and resolution for XL-DOT measurements.

## Data availability

Information needed to evaluate the presented conclusions are provided in the manuscript and the Supplementary Information. The raw data and reconstructed projections can be accessed at https://doi.org/10.5281/zenodo.14258649 (ref. 90) or obtained from the corresponding authors.

## Code availability

Tomographic reconstruction and analysis codes can be accessed at https://doi.org/10.5281/zenodo.14258649 (ref. 90) or obtained from the corresponding authors. General acquisition and reconstruction codes can be accessed at https://www.psi.ch/en/sls/csaxs/software.

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

**Acknowledgements** General: tomography experiments were performed at the cSAXS beamline of the Swiss Light Source (SLS), Paul Scherrer Institute (PSI). We thank X. Donath and P. Zimmerman for technical support. Electron-microscopy work was performed at the Scientific Center for Optical and Electron Microscopy (ScopeM) at ETH Zürich and the Electron Microscopy Facility at PSI, with the assistance of E. A. Mueller Gubler, J. Reuteler and A. G. Bitterman. We would also like to thank I. Robinson for the insightful discussion. Funding: the research leading to these results has received financing from the Swiss National Science Foundation (SNSF), with project numbers 200021_192162 (A.A.), 200021_196898 and PZ00P2_179886 (Z.G. and J.I.), the Max Planck Society Lise Meitner Excellence Program (C.D.), the European Research Council (ERC) under ERC Starting Grant no. 3DNANOQUANT 101116043 and the EU's Horizon 2020 research and innovation programme under the Marie Skłodowska-Curie grant agreement no. 884104 (PSI-FELLOW-III-3i) (C.A.).

**Author contributions** J.I., C.D. and M.G.-S. conceived the study. V.S. and M.H. designed, set up and aligned the polarization stage. J.I. prepared the sample. A.A., M.H., M.G.-S., C.A., V.S., C.D. and J.I. performed PXCT experiments. A.A. and C.D. developed the orientation tomogram reconstruction algorithm. A.A. performed PXCT with support from Z.G. and X-ray linear dichroic orientation tomography reconstructions with support from C.D. A.A. analysed the data, with the help of C.D., M.G.-S. and J.I. A.A., J.I., C.D., V.S. and L.J.H. wrote the manuscript, with contributions from all authors.

**Funding** Open access funding provided by Max Planck Society.

**Competing interests** The authors declare no competing interests.

**Additional information**
**Correspondence and requests for materials** should be addressed to Andreas Apseros, Valerio Scagnoli or Claire Donnelly.
