## [Peer Review file · Nature]

X-ray Linear Dichroic Tomography of Crystallographic and Topological Defects

Corresponding Author: Dr Claire Donnelly

Version 0:

Reviewer comments:

Referee #1

(Remarks to the Author)

This is a very thorough and novel work that deserves publication in Nature. The novelty in developing this new modality of 3D structural microscopy is exceptional. The demonstration on a porous polycrystalline sample V₂O₅ undoubtedly demonstrates the potential of the method to resolve structure within certain complex materials to levels of detail never before achieved. The authors achieve this by marrying the high resolving power of ptycho-tomography with the phase contrast sensitivity of x-ray linear dichroism to asymmetric molecular arrangements. This was done with an innovative experimental design that enabled the vector orientation of the asymmetric apical bond of the molecular unit cell to be resolved. The experimental design (involving multiple polarization angles and tomographic axes or rotation) necessitated the development of a problem-specific image reconstruction approach. The authors achieve this by solving an inverse problem by iterative gradient descent, wherein the complete set of 2D projections of the sample measured with dichroic x-ray phase contrast is compared with the equivalent set emulated from a proposed model of the 3D distribution of c-axis orientation via an elegant forward model. This represents another major and novel aspect of this work. The successfully reconstructed XL-DOT tomogram was able to provide rich information about the inner structure of the sintered V₂O₅ microns-sized sample, including the existence of polymeric residue and voids, identification of individual grains segmented by c-axis orientation, texture within those grains, details on the nature of grain boundaries, and the spatial evolution of topological defects. As the authors rightly point out, such differentiation of grains by crystallographic orientation is not possible by other methods at the tens of nanometer spatial scales demonstrated here. The significance of this work very clearly lies in the fact that gaining nanoscale crystal-lattice orientation information in within polycrystals nondestructively represents a very important step forward in developing structure/property/function relationships in polycrystalline materials, which are ubiquitous in engineering and technology.

In my view, the main weakness in this work is the fact that the contrast mechanism of x-ray linear dichroism, because it is effective only near elemental absorption edges and only for certain materials, may hamper the impact and reach of this method regarding studying materials responses under working conditions, which is one of the claims made by the authors. I recommend that the caveats of the method be more clearly discussed, especially with regard to prospects of applying the method more broadly. Factors that should be put more clearly into context include:

- the fact that K-edge energies of many of the transition metals do not afford particularly high x-ray penetration depths, making it difficult to design operando experiments;
- the need for multi-axial tomographic measurements raises a similar concern in terms of measurement time and compatibility with sample environments;
- an aspect of lattice orientation determination that is altogether not discussed in this manuscript is that the crystallographic orientation of the a and b axes of the unit cell about the c axis cannot be determined. It is important to discuss this, as many issues of materials phenomenology, such descriptions of dislocation slip systems across grain boundaries in structural materials for example, require the full orientation matrix at the grain level.
- many classes of materials do not show sufficient linear dichroism and cannot be studied with XL-DOT.

These aspects are to some degree covered in the work, but in the closing paragraphs of the manuscript the authors seem to champion the upsides of this method without a very balanced view of the limitations, some of which pose a fundamental challenge.

A technical question also arose in my reading of this manuscript that the authors should address: What is minimum number of tilt angles / polarization states needed for a stable reconstruction? The current data set likely overdetermined the image

reconstruction problem to solve for c-axis orientation in 3D. Making viable XL-DOT reconstructions with fewer scans will enable more flexible design of operando measurements and higher throughput for capturing evolving sample states.

My main critique of the work, namely that the context of this new method for broader materials characterization be presented in a more balanced light, in no way detracts from the considerable novelty of the very impressive work presented in this manuscript.

The validity of the approach, the quality of the data, and the quality of presentation are sound. The robustness of this method and uncertainties in the figures of merit are well documented and thoroughly discussed. The references are appropriate and extensive. The clarity of presentation is very good.

Referee #2

(Remarks to the Author)

Andreas Apseros et al. present an exciting study on the possibility to use dichroic signal produced by an anisotropic crystal to retrieve the 3D crystalline micro and nano structure of an extended sample. The new method they introduce, named X-ray linear dichroic orientation tomography (XL-DOT), is based on a ptychography tomographic treatment of the dichroic contrast associated to the orientation-dependent answer of a crystal under different linearly polarized illuminations. This is a fully original use of the x-ray dichroic contrast of a crystal, where its dependence on the 3D orientation of the crystal c-axis is incorporated into the ptychography signal model.

The paper is meticulously written, with considerable effort dedicated to ensuring accessibility to a wide audience. The shown demonstration is fully convincing, based on the examination of a purposely prepared assembly of vanadium pentoxide crystalline particles via a sintering method. This choice is clever, as sintering induces the development of crystalline defects that propagates over significant length scales. The subsequent 3D analysis of topological crystalline defects and their interactions highlights the power of the method and its capability to address crystalline defects analysis in a novel and elegant manner.

Up to this point, a few different strategies were proposed to address the question of crystalline distortion characterization. On the one hand, Bragg coherent diffraction imaging based approaches [Ulvestad et al., Science 2015, Li et al. Nature Commun 2021 and ref. 10 of the manuscript] collect the coherent diffraction signal measured in the vicinity of a Bragg peak to extract the crystalline displacement, from which lattice rotations and dislocations can be identified and characterized. They provide a highly sensitive and highly spatially resolved description of the crystalline structure, but within a limited crystalline distortion range (typical lattice mis-orientation below 2-3°). On the other hand, diffraction contrast tomography, proposed in the late 2000 [Ludwig et al., J. Appl. Cryst. 2008 and Ref 15 of the manuscript], is able to probe polycrystalline material presenting fully mis-oriented grains. However, the sensitivity of the method prevents the detailed analysis of the intra-grain distortion analysis. Another relevant approach, which the authors did not refer to, is dark-field microscopy [see e.g., Simons et al., Nature Commun. 2015], where an optical element is placed behind the sample, along a Bragg diffraction peak signal produced by one of the crystallites. The extensive scanning process grants access to a 3D map of the crystalline axis orientation, offering high sensitivity to crystalline distortion. However, the spatial resolution is constrained by the limitations of the X-ray optics.

With XL-DOT, the portfolio of crystalline microscopy is enriched by a method, which is more sensitive than diffraction contrast tomography, which overcomes the thickness limit of coherent diffraction imaging microscopy and which provides a better spatial resolution than the one imposed by the optical element of dark-field microscopy. Therefore, one can expect that new questions in material science at the nanoscale will be soon addressed, like the mechanism responsible for platelet stacking in columnar nacre.

For these reasons, I fully support the publication of this article in Nature. However, several issues should be addressed in order to improve the understanding of the potential impact of the method and its domain of application, which I think the manuscript fails to convey. I detailed them below:

I - Detailed analysis of the limits of the method:

In its present form, a detailed analysis of the method limits is missing. Angular sensitivity and spatial resolution are discussed in details, and a series of tests and analysis are provided to support the conclusions. They were highly appreciated.

- Regarding spatial resolution, several analyses are included to discuss in details this question. This is totally fair and needed. I would appreciate that the spatial resolution discussion also includes an analysis on a smaller crystallite, similarly to what was done in Figure S8. This would allow evaluating whether the spatial resolution is a global parameter or whether it depends on the size of the scattering object.

- The energy dependence is not really discussed. In particular, how accurate should be the energy positioning with respect to the chosen edge? What happens if the material is composed of different compounds (or even polymorphs) with the same edge but different charge states / slightly varying energy profile at resonance? (for instance if the material is composed of aragonite and calcite sub-structures).

- Regarding the dichroic contrast induced by the crystalline orientation: I understand that delivering the reference spectra presented in Figure S4 is crucial. It allows providing the value of the factor A in equation S1 (it fixes the amplitude of the dichroic contrast for the two extreme orientations). However, delivering these spectra requires accurate crystal orientation positioning and intensity normalization. How is it done in practice?

Further more, only samples that present an optical anisotropic behaviour in the x-ray regime can be investigated. How large

is this class of materials?

- Temporal resolution : XL-DOT is expected to be compatible with operando approaches (see e.g., lines 57 and 286-292 of the manuscript). However, the acquisition procedure is rather time consuming. What was the total acquisition time of the presented result (and can you add the information in the manuscript)? How can the acquisition time be reduced in the future and which reducing factor to expect? What temporal resolution could be targeted in fine?

II - Comparison to other methods:

In order to provide a fair assessment of XL-DOT with respect to the other crystalline microscopy methods, I believe a dedicated paragraph is needed in the discussion part. In its present version, this question is briefly addressed in a paragraph at the end of the manuscript (lines 280-285). This part should be improved, with clear references and positioning with respect to x-ray dark-field contrast microscopy, x-ray diffraction contrast tomography, Bragg CDI and Bragg ptychography. In particular, the strain seems to be not accessible with XL-DOT, is it correct?

III - Other major remarks:

- Please provide the definition of the topological charge for non-specialist readers.
- Line 294: what do you mean by orientation of amorphous material? By definition, an amorphous material does not have long range ordering.
- Line 309: How did the FIB damage the sample? Can you briefly describe?
- Density: how was the electron density of the defect-rich V2O5 established in Table S1? In Figure S9, it seems there is a shift between the mark '(ii)' corresponding to the established density and the maximum of the electron density distribution. Is it significant?

IV Additional minor remarks

- Typos on line 505 and 508, on the density units

Referee #3

(Remarks to the Author)

Apseros et al implement linear dichroic ptychographic tomography with very high resolution in an impressive experimental tour de force. The work is technically impressive and interesting but for the reasons listed below, I am not convinced about the general applicability, uniqueness and impact of the method. Therefore, I cannot support publication of this paper in Nature.

The authors state that no other methods except TEM give 3D orientation information on the nanoscale. However, dark field X-ray microscopy (Simons, H., King, A., Ludwig, W. et al. Dark-field X-ray microscopy for multiscale structural characterization. Nat Commun 6, 6098 (2015)) and Bragg ptychography (Mastropietro, F., Godard, P., Burghammer, M. et al. Revealing crystalline domains in a mollusc shell single-crystalline prism. Nature Mater 16, 946–952 (2017)) give such information at similar resolution as established in the present manuscript and additionally directly provide information on e.g. strain. Neither method is discussed by the authors, which is a significant problem.

The authors indicate that the proposed method is widely applicable to map orientation. However, orientation is obtained by proxy in the proposed approach: the authors map the X-ray linear dichroism, which is related to orientation but not in a manner that – in general – can be simply linked to the signal. This is explored extensively in the PEEM work of PUPA Gilbert who demonstrated the need to measure control samples to link orientation and dichroism. In the present example, the authors make use of the special symmetry and structure in V2O5 that defines the relationship between orientation and dichroism. However, such a relationship does not exist in general. This limits the scope of the proposed method and is not even mentioned in the manuscript by the authors. The claim on page 2 that the method can be used to map amorphous materials in 3D does not seem warranted.

The method, as I understand it, requires the measurement of multiple ptychography datasets (8!). The current measurement involved a dose of a GGy. In practice, this means that there will certainly be beam damage in many sample classes, e.g. polymers, which is already a limiting factor in many 'simple' ptychography experiments. Again, this is an issue that is not really discussed but that in practice will limit the methods applicability. This certainly applies to the rather broad sweeping concluding remarks on page 13: "This opens the door... bone". The authors would have to demonstrate some of these capabilities or remove these claims.

On page 15: "Although XL-DOT..." does not mention the PEEM work of PUPA Gilbert (which is cited later), which certainly merits mention here – it is 2D but only requires a simple polished surface.

On page 18, the authors mention dose, but the actual time the measurements took is never mentioned. Given the claims that the method should be applicable in situ/operando, the real measurement time is important and should be given.

In the section "Measurement Error Estimation" on page 18, the authors report the average measure phase shift to be $5.8E-6$ with a standard deviation of $1.3E-4$ rad. This would correspond to a standard deviation that is 1.5 orders of magnitude larger than the signal – please explain?

Taken together, the manuscript presents a technical tour de force but does in my opinion not demonstrate a widely applicable advance or novel insights into an important material. The method has rather significant limitations that the authors downplay or ignore, and relevant competing methods are not discussed. I therefore cannot support publication of this paper.

Version 1:

Reviewer comments:

Referee #1

(Remarks to the Author)

The authors have sufficiently addressed all of my comments in their reply and in the changes made to the manuscript. The effort to add new figures and context greatly improves this impressive work.

Referee #2

(Remarks to the Author)

I am fully satisfied with the revised version proposed by Andreas Apseros et al. My questions have been considered with great care. The detailed answers are fully convincing. The modifications made to the manuscript fully take into account my remarks. The manuscript is clearly improved. It is now more balanced regarding its positioning with respect to the scientific framework and other methodologies and more impactful regarding the class of (crystalline and non-crystalline) materials the proposed method is able to address. I think it can be accepted for publication and I strongly support its acceptance in Nature journal.

Referee #3

(Remarks to the Author)

The revised manuscript of Apseros et al. is significantly improved compared to the first version of the manuscript. I really appreciate how the authors now address limitations of XL-DOT and inclusion of other methods. This has resulted in a much more balanced manuscript. I am therefore now prepared to possibly support acceptance of the manuscript in Nature provided suitable responses to a few additional concerns are provided:

The authors have in their response compared reconstructions with a varying number of tilts (new Figure S14). This is most welcome and an excellent addition to the good existing analysis of reconstruction fidelity. However, I don't follow their conclusion that the 2-tilt reconstruction "there appears to be an overall agreement in terms of the reconstructed orientation with the four-tilt reconstruction." While it is certainly true that a proportion of the orientation map coincides, there are important differences, some of which I have encircled in the attached file. If one only had access to the 2-tilt reconstruction, these would be interpreted as smaller grains or similar, possibly resulting in erroneous conclusion on the properties of the sample. My reading of figure S14 is rather that it demonstrates that at least 3 tilts are needed for appropriate reconstruction of orientation. This does not detract from XL-DOT as a technique. But it leads to a question: do the authors foresee a possibility to also extract a local confidence map, where one could evaluate the 'noise' in the reconstruction? This would be a most interesting addition, but I fully understand if it is beyond the scope of the present work. However, I do think that the presentation of Figure S14 should be changed – in my reading it does not demonstrate that two tilts are enough, rather that four are much better. The authors should either do a full (3D) quantitative analysis of the two vs four tilt reconstruction or simply conclude that at present four tilts are needed but that further work on acquisition schemes may reduce the number of required tilts. I really think that the authors (and XL-DOT) are better served by being a bit conservative here.

In line 285-286, I suggest adding also Bragg ptychography so that the phrase becomes "...of nanoscale samples with BCDI, Bragg ptychography, photoemission electron microscopy (PEEM)..." for balance in presentation of the techniques.

Line 302-305: the authors retain their statement that XL-DOT will be suitable for orientation mapping of biominerals, now writing "...both the organic and inorganic parts...". The cited example in reference 40 deals with PEEM studies of Ca dichroism while reference 46 deals with optical LD of polymers, as I read it (As I understand reference 46, wavelengths 170-320 nm). For the organic component, the cited preprint, reference 46, studies optical LD. In my reading, the authors state that XL-DOT can be applied in 3D on samples of e.g. bone using the optical dichroism – or do they envision hard X-ray LD from HCNO-containing polymers to be strong enough to make the experiment successful? Maybe I am simply not understanding what the authors intend to say, but please provide an estimate of the transmission possible with a realistic biomineral sample to test feasibility of LD measurements of the polymers in several micron sized biomineral specimens (bone as many biominerals is white due to very significant light scattering that limits transmission in the visual domain and proteins absorb strongly around 280 nm). What sample sizes do the authors envision will be measurable? Do the authors have an estimate of what 'concentration' the target element should have to allow XL-DOT measurements? In many biocomposites, the concentration of the target elements, e.g. Ca, are significantly lower than in the present example of V2O5.

Concerning the discussion (lines 309-320) on in operando: it is in principle true that almost any experimental technique can study sufficiently slow dynamics. The relevant question becomes whether the time scales that can be studied are relevant for the stated materials classes. Even given the foreseen improvements in technology, XL-DOT will – as I read the response from the authors – remain a relatively slow (hour-scale) technique. This does not make it bad, but it does place limits on the

applicability for in operando studies. I urge the authors to either acknowledge this (stating the 'slow' can be followed or that there will a time resolution limit) or (in my view the better option) leave out the section on in operando measurements. The information obtained in XL-DOT is interesting enough on its own without this perspective outlook on in operando studies.

Author Rebuttals to Initial Comments:

We thank the referees and editorial staff for their detailed, constructive, and positive reviews of our manuscript, which have led to significant improvements in the quality and presentation of our work.

As suggested by the referees, we have added a more balanced discussion, highlighting the capabilities and limitations of XL-DOT. We have also added discussions and additional figures clarifying: 1) the question of compatibility with operando measurements, 2) the sample diameter - photon energy dependency, 3) deposited dose, 4) current acquisition speed and 5) the multi-axial sampling requirement, more specifically, the prospect for simpler measurement setups. Lastly, to appeal to a broader scientific audience and to make the manuscript more accessible we have added paragraphs providing, first an overview of the range of materials / science cases that can potentially be studied with XL-DOT, and subsequently a discussion of how XL-DOT compares and complements the current state-of-the-art microstructural characterisation techniques.

Provided below is a point-by-point response to the referees' comments. The comments are presented in black, the responses in blue. Changes to the manuscript, including additional or updated figures, are quoted below the respective replies. Changes to the manuscript text are highlighted in a bold font.

Referee #1 (Remarks to the Author):

This is a very thorough and novel work that deserves publication in Nature. The novelty in developing this new modality of 3D structural microscopy is exceptional. The demonstration on a porous polycrystalline sample V2O5 undoubtedly demonstrates the potential of the method to resolve structure within certain complex materials to levels of detail never before achieved. The authors achieve this by marrying the high resolving power of ptycho-tomography with the phase contrast sensitivity of x-ray linear dichroism to asymmetric molecular arrangements. This was done with an innovative experimental design that enabled the vector orientation of the asymmetric apical bond of the molecular unit cell to be resolved. The experimental design (involving multiple polarisation angles and tomographic axes or rotation) necessitated the development of a problem-specific image reconstruction approach. The authors achieve this by solving an inverse problem by iterative gradient descent, wherein the complete set of 2D projections of the sample measured with dichroic x-ray phase contrast is compared with the equivalent set emulated from a proposed model of the 3D distribution of c-axis orientation via an elegant forward model. This represents another major and novel aspect of this work. The successfully reconstructed XL-DOT tomogram was able to provide rich information about the inner structure of the sintered V2O5 microns-sized sample, including the existence of polymeric residue and voids, identification of individual grains segmented by c-axis orientation, texture within those grains, details on the nature of grain boundaries, and the spatial evolution of topological defects. As the authors rightly point out, such differentiation of grains by crystallographic orientation is not possible by other methods at the tens of nanometer spatial scales demonstrated here. The significance of this work very clearly lies in the fact that gaining nanoscale crystal-lattice orientation information in within polycrystals nondestructively represents a very important step forward in developing structure/property/function relationships in polycrystalline materials, which are ubiquitous in engineering and technology.

We would like to thank the referee for their detailed review and for the time and effort taken to thoroughly examine the manuscript and provide constructive feedback. We are glad to receive their positive review, as well as their recognition of the new technique and its potential.

In my view, the main weakness in this work is the fact that the contrast mechanism of x-ray linear dichroism, because it is effective only near elemental absorption edges and only for certain materials,

may hamper the impact and reach of this method regarding studying materials responses under working conditions, which is one of the claims made by the authors. I recommend that the caveats of the method be more clearly discussed, especially with regard to prospects of applying the method more broadly. Factors that should be put more clearly into context include:

- the fact that K-edge energies of many of the transition metals do not afford particularly high x-ray penetration depths, making it difficult to design operando experiments.
- the need for multi-axial tomographic measurements raises a similar concern in terms of measurement time and compatibility with sample environments;
- an aspect of lattice orientation determination that is altogether not discussed in this manuscript is that the crystallographic orientation of the a and b axes of the unit cell about the c axis cannot be determined. It is important to discuss this, as many issues of materials phenomenology, such as descriptions of dislocation slip systems across grain boundaries in structural materials for example, require the full orientation matrix at the grain level;
- many classes of materials do not show sufficient linear dichroism and cannot be studied with XL-DOT.

These aspects are to some degree covered in the work, but in the closing paragraphs of the manuscript the authors seem to champion the upsides of this method without a very balanced view of the limitations, some of which pose a fundamental challenge.

We thank the referee for their suggestion to include a more balanced discussion about the advantages and limitations of the presented technique. The discussion section has been changed accordingly. A detailed, point-by-point response and resulting manuscript changes to the specific points raised by the referee are provided here:

1. the fact that K-edge energies of many of the transition metals do not afford particularly high x-ray penetration depths, making it difficult to design operando experiments;

We agree with the reviewer that the requirement to perform measurements at linear dichroism exhibiting features of an absorption edge (K-, L-, or M-), i.e. at a pre-defined photon energy, could lead to an upper sample diameter limit. However, we don't think that this limitation prevents operando measurements for the following reasons:

a) The X-ray penetration depth or transmissivity of transition metals at the K-edge energy is ~10 μm . Transition-metal-rich materials, and first operando XL-DOT targets, such as catalysts, cathodes, ferroelectrics, as well as biominerals and concrete samples exhibit a greater transmissivity, due to their composite or porous nature [I. D. Gonzalez-Jimenez et al. *Angewandte Chemie* **124** 12152 (2012)]. This will allow for 10's to 100's of μm s to be studied.

b) Although one may think that such sample diameters may be a limiting factor, they are currently the target diameter for (operando) nanotomography, with a number of operando measurements having already been demonstrated for such sample sizes [M. Holler et al. *J. Synchrotron Rad.* **29** 1223 (2022), C. Donnelly et al. *Nat. Nano.* **15** 356 (2020), A. Ulvestad et al. *Nano Lett.* **14** 5123 (2014), S.M. Bak et al. *NPG Asia Mater* **10** 563 (2018)]. As a result, we believe it will be possible to combine XL-DOT with such operando measurements in the near future, making it possible to image the evolution of 3D orientation fields.

We note that, even taking a conservative estimate of the viable sample sizes of $\sim 10\ \mu\text{m}$, the use of XL-DOT results in an order of magnitude increase in sample size than is currently possible with techniques such as electron microscopy or Bragg CDI.

To clarify this situation further we added a further subsection to the Methods, lines 590 to 600:

“Sample diameter and photon energy resolution considerations: As linear dichroic phenomena occur near absorption edges or resonant X-ray energies, the x-ray penetration depth at these energies determines the sample diameter that can be investigated with XL-DOT. For most materials, it is the penetration depth at the X-ray energy of the probed chemical element that sets an upper limit on the sample diameter. Taking pure transition metals as an example, this imposes a typical upper sample limit of around $10\ \mu\text{m}$. Transition-metal-rich functional materials such as catalyst bodies, cathode materials, ferroelectrics, biominerals and concrete, which are also of interest for XL-DOT measurements, exhibit a substantially larger upper sample size limit due to their internal porosity or composite nature. For instance, a $100\ \mu\text{m}$ thick V_2O_5 sample transmits around 10% of the incident beam in the pre-edge region (henke.lbl.gov). 3D or nanotomography measurements of such sample diameters are increasingly typical for operando measurements. [M. Holler et al. J. Synchrotron Rad. 29 1223 (2022), C. Donnelly et al. Nat. Nano. 15 356 (2020), A. Ulvestad et al. Nano Lett. 14 5123 (2014), S.M. Bak et al. NPG Asia Mater 10 563 (2018)]”

2. the need for multi-axial tomographic measurements raises a similar concern in terms of measurement time and compatibility with sample environments;

We understand the concern of the referee about the time and complexity of multi-axial tilt measurements. Fortunately, there are opportunities to reduce the complexity of the measurement.

a) The measurements presented in our manuscript are to a certain degree “oversampled” in terms of sample tilts and or linear polarisation states. This “oversampling”, while improving the signal-to-noise ratio and the reconstruction quality, was deemed necessary in these first experiments to make sure the data could be reconstructed successfully. To explore the effects of the number of tilt angles, we have now performed extensive numerical simulations of XL-DOT using a polycrystalline test object (see Figure R1a). The numerical simulations of XL-DOT with multiple tilt axes, and horizontal and vertical linear polarisation, show that four tilts or more yield optimal results (see Figure R1b), although a robust reconstruction can be obtained with as few as two tilts. To test whether such a two-tilt measurement would indeed provide a robust reconstruction, we performed a 3D reconstruction of the crystal orientation using data for V_2O_5 from only two tilt axes. We observe that, in general, a robust reconstruction is indeed achieved, albeit with a slightly lower spatial resolution and increased angular error (see Figure R1c,d). This combination of numerical simulations and additional data analysis indicates that, although we have used four tilt axes in this demonstration, a robust reconstruction can be obtained with a less complex setup with fewer tilt axes.

Figure R1: a) Slice obtained from the 3D domain structure used in simulations. b) Plot of angular error obtained from numerical simulations of the XL-DOT technique against number of tilt axes. The shaded areas represent the 5th to 95th percentile errors (light grey) and 25th to 75th percentile errors (dark grey). The red data points (line is a guide to the eye) represent the mean angular error. Slices through the XL-DOT reconstructed sample volume obtained from b) two tilt axes and c) four tilt axes reconstructions.

b) Our simulations (to be published in a separate, follow-up manuscript), and as suggested by others [M. A. Marcus Opt. Express **30** 39445 (2022), Y. H. Lo et al. P. Natl. A. Sci. **118** (2021)], indicate further that the reduction in information with the fewer number of tilt axes can be compensated by more incident x-ray polarisations, e.g. the measurement of additional tomograms at a fixed tilt angle, with for example a linear x-ray polarisation of 30°, 45° and 75°. This offers a route to fewer or even single tilt axis measurements with high quality reconstructions of the 3D orientation distribution and the integration of XL-DOT in already existing operando nanotomography setups [M. Holler et al. J. Synchrotron Rad. **29** 1223 (2022)].

We are currently not able to measure with arbitrary X-ray polarisation due to current infrastructural limitations at the cSAXS beamline of the Swiss Light Source, i.e. the absence of an undulator able to change the linear polarisation state of the incident X-ray illumination [S. Sasaki Nucl. Instrum. Meth. A **347** 83 (1994)]. Undulators with this capability are increasingly common and will make XL-DOT measurements substantially more accessible and easier to implement. Advanced diamond phase retarders could also be used [V. Scagnoli et al. J. Synch Rad **16** (2009)].

c) Lastly, it is important to note that a key step towards operando measurement for circular dichroic 3D vector magnetic imaging was made by the implementation of the laminography geometry, which reduces the number of rotation axes directly to one [C. Donnelly et al. Nat. Nano. **15** 356 (2020), Witte et al. Nano. Lett. **20** 1305 (2020)]. Although not shown here, our simulations also suggest that the laminography geometry will make single rotation axis XL-DOT measurements possible, with such measurements playing an important role for future implementations of magnetic and electrochemical operando measurements.

We have included a comparison of the four-tilt and two-tilt reconstructions in a new Supplementary Figure, Figure S14:

“Supplementary Figure S14: Impact of the number of tilts on the reconstruction quality. a) Plot of angular error obtained from numerical simulations of the XL-DOT technique against number of tilt axes. The shaded areas represent the 5th to 95th percentile errors (light grey) and 25th to 75th percentile errors (dark grey). The red data points (line is a guide to the eye) represent the mean angular error. From the plot it is evident that four tilts or more yield optimal results, although a robust reconstruction can be obtained from as few as two tilts. Slices through the XL-DOT reconstructed sample volume obtained from b) two-tilt and c) four-tilt reconstructions. Although the reconstruction with only two tilt axes has an increased angular uncertainty and lower spatial resolution, there appears to be an overall agreement in terms of the reconstructed orientation with the four-tilt reconstruction.”

We have also added a further subsection to the methods on lines 502-508 on page 19 as follows

“*Multi-axis tomography*: To obtain a first estimation of how many sample tilts and linear polarisation states are necessary for a robust XL-DOT reconstruction, we performed a series of numerical simulations and tomographic reconstructions with fewer sample tilt axes (Figure S14). Robust reconstructions can be obtained with as little as two tilt axes using LV and LH polarisations only. Both our simulations (not shown) and recent literature [M. A. Marcus Opt. Express 30 (2022)] and Y. H. Lo et al. [Proc. Natl. Acad. Sci. U.S.A. 118 (2021)] further indicate that the multiple tilt axes can be replaced by measurements with additional X-ray polarisations [S. Sasaki Nucl. Instrum. Meth. A 347 83 (1994)]. Similar results can also be achieved using laminography. [M. Holler et al. J Synchrotron Rad 27, 730 (2020), Donnelly, C., et al. Nat. Nano. 15 (2020)]. This offers a route to using fewer tilt axes or even single tilt axis measurements.”

We have also updated the discussion on lines 318-322 to address these points:

“[...] Nevertheless, we note that a robust reconstruction can also be obtained with only two tilt axes (Figure S14), reducing the complexity of the experimental setup. Moreover, we envisage that the use of alternative geometries [M. Holler et al. J Synchrotron Rad 27, 730 (2020), Witte et al. Nano. Lett. 20 1305 (2020), M. A. Marcus Opt. Express 30 39445 (2022)], which only require one tilt axis, will also allow for an easier implementation of *operando* imaging as demonstrated for vector magnetic imaging [C. Donnelly et al. Nat. Nano. 15 (2020)].”

3. an aspect of lattice orientation determination that is altogether not discussed in this manuscript is that the crystallographic orientation of the a and b axes of the unit cell about the c axis cannot be determined. It is important to discuss this, as many issues of materials phenomenology, such as descriptions of dislocation slip systems across grain boundaries in structural materials for example, require the full orientation matrix at the grain level.

We thank the reviewer for the suggestion to clarify this point. It is indeed the case that with a single-energy XL-DOT measurement, the crystallographic orientation of the secondary axis of the unit cell about the probe axis is undetermined.

Addressing this shortcoming, alongside achieving higher spatial resolution, represents the next major development step. The crystallographic orientation of the *a* and *b* axes can, for example, be obtained by implementing a second detector to measure low-resolution diffraction patterns. The corresponding low-resolution X-ray diffraction tomogram (with a resolution equivalent to the scanning probe size of μms) can be used to refine/constrain the XL-DOT measurement, and with this determine the crystallographic orientation of the secondary axis. A second option, currently under consideration, is the combination of XL-DOT with sparse X-ray transmission near-edge spectroscopic tomography [Z. Gao et al. *Sci. Adv.* **7** (2021), N. Ishiguro and Y. Takahashi *J. Appl. Cryst.* **55** 929 (2022)]. Since secondary orientations/anisotropies can potentially be probed at different energies in the vicinity of the absorption edge, we envisage that the resulting spectroscopic XL-DOT datasets could provide the desired information in tandem with a chemical characterisation of the material and defects. These options involve a substantial financial investment and/or development time (multiple years). As a result, they are not within the scope of the present manuscript.

To make this prospect clear, we have added the following to Supplementary Note 3 on lines 907 to 917 as well as a pointer to this note in the main text when mapping of the *c*-axis orientation is first mentioned, on line 80:

“Since we performed XL-DOT at a single X-ray energy, only the *c*-axis orientation of V_2O_5 was probed, and the relative orientation of the crystallographic *ab* axes of the unit cells is undetermined. While this is sufficient to address several aspects of microstructural characterisation, the orientation determination of the secondary axes would be very helpful, for example, for the in-depth characterisation of crystallography defects such as dislocations [S. Maddali et al. *npj Comput Mater* **9, 77 (2023)]. For materials where the axes $a \neq b \neq c$, the full orientation tensor, could, for example, be obtained through the implementation of a second detector that simultaneously collects a low-resolution diffraction tomogram to refine the high-resolution XL-DOT dataset. For selected materials, a second option involves the collection of spectroscopic XL-DOT data across an absorption edge. Such a spectroscopic measurement could benefit from sparsity as performed for x-ray transmission near-edge spectroscopic tomography [Z. Gao et al. *Sci. Adv.* **7** (2021), N. Ishiguro and Y. Takahashi *J. Appl. Cryst.* **55** 929 (2022)]. The resulting spectroscopic XL-DOT datasets, which probe secondary anisotropies, would provide the desired orientation of the *ab* axes in tandem with a chemical characterisation of the defects.”**

4. many classes of materials do not show sufficient linear dichroism and cannot be studied with XL-DOT.

While not all materials exhibit linear dichroism – for example materials with isotropic properties, where the electronic or structural environment is the same in all directions – all non-cubic crystalline, magnetic and low-dimensional (e.g. graphene) materials, as well as selected coordination complexes and molecular arrangements/networks, will exhibit linear dichroism to some degree. As such, our technique is applicable to a wide variety of materials [J. Stöhr “The Nature of X-Rays and Their Interactions with Matter” ch.7, S. Polisetty et al. *J. Phys.: Condens. Matter* **24** (2012), H. Ade and B. Hsiao *Science* **262** 1427(1993), C. Jansing et al. *J. Phys.: Conf. Ser.* **712** 012031 (2016)].

The development of XL-DOT will be of particular relevance, together with future operando capabilities, to transition metal oxides, sulphides, carbonates and phosphates, which represent a substantial fraction of heterogeneous catalysts, energy storage cathode and solid electrolyte materials, biominerals and cementitious materials with immediate societal impact. Additionally, as magnetic

anisotropy is a source of linear dichroism, XL-DOT can be employed to map the real space Neel vector in antiferromagnets in 3D, which until now has not been possible.

Undoubtedly, for some of these materials, the linear dichroism is relatively weak and, while XL-DOT is compatible with other imaging modalities, the utilization of coherent imaging techniques such as ptychography, which provides access to the phase contrast and improved signal-to-noise ratio, facilitates the detection of even weak linear dichroic signals [Z. Gao et al. Chem. Commun. 56, 13373 (2020), Monaco, F. Mater. Charact. 187 111834 (2022)]. Indeed, ptychography has already been shown to be particularly effective for imaging weak signals, such as the x-ray magnetic circular dichroism exhibited by ferromagnets [C. Donnelly et al Phys. Rev. B **94** 064421 (2016)]. We envisage that, with the increase in coherent flux at 4th generation synchrotron radiation sources, even weaker dichroic signals will become reliably measurable.

Beyond anisotropic materials, we envisage that, upon substantial increases in spatial resolution, we will be able to use XL-DOT to probe the local anisotropies emergent from crystallographic defects such as dislocations, even in isotropic, cubic materials.

To clarify this point in the main text, we have included the following on lines 301-304 and 323-328:

“In addition, beyond mapping the orientation of non-cubic crystalline systems by exploiting the linear dichroism that they exhibit, the dichroic contrast mechanism can be used to map the orientation of the order parameter in non-crystalline materials (such as some ferromagnets) that until now could not be characterised with diffraction-based techniques alone.”

“Not only will this dramatically speed up measurements, it will also bring the spatial resolution down to the order of ten nanometres, providing a way to map smaller point and line defects, such as dislocations. The increase in coherent flux will make even weaker dichroic signals detectable, making it possible to map the associated orientation of the order parameter of an even wider variety of materials in 3D.”

5. A technical question also arose in my reading of this manuscript that the authors should address: What is minimum number of tilt angles / polarization states needed for a stable reconstruction? The current data set likely overdetermined the image reconstruction problem to solve for c-axis orientation in 3D. Making viable XL-DOT reconstructions with fewer scans will enable more flexible design of operando measurements and higher throughput for capturing evolving sample states.

We thank the reviewer for this very pertinent question. The number of tilt axes/polarisation states required to obtain a robust reconstruction of the orientation field is indeed a key point to consider for future applications and operando measurements. In short, we have identified that a valid reconstruction can be obtained by measuring a minimum of two tilt axes, albeit at a lower spatial resolution and with some orientation uncertainty. A detailed reply and manuscript changes regarding the necessary number of tomographic tilt axes are provided under Point 2, on Pages 3-5 of this document.

My main critique of the work, namely that the context of this new method for broader materials characterization be presented in a more balanced light, in no way detracts from the considerable novelty of the very impressive work presented in this manuscript.

The validity of the approach, the quality of the data, and the quality of presentation are sound. The robustness of this method and uncertainties in the figures of merit are well documented and thoroughly discussed. The references are appropriate and extensive. The clarity of presentation is very good.

We thank the referee for their positive assessment of our manuscript, as well as their constructive feedback, which we believe has led to a much more balanced discussion of the results and a significant improvement in the quality of the manuscript.

Referee #2 (Remarks to the Author):

Andreas Apseros et al. present an exciting study on the possibility to use dichroic signal produced by an anisotropic crystal to retrieve the 3D crystalline micro and nano structure of an extended sample. The new method they introduce, named X-ray linear dichroic orientation tomography (XL-DOT), is based on a ptychography tomographic treatment of the dichroic contrast associated to the orientation-dependent answer of a crystal under different linearly polarized illuminations. This is a fully original use of the x-ray dichroic contrast of a crystal, where its dependence on the 3D orientation of the crystal c-axis is incorporated into the ptychography signal model.

The paper is meticulously written, with considerable effort dedicated to ensuring accessibility to a wide audience. The shown demonstration is fully convincing, based on the examination of a purposely prepared assembly of vanadium pentoxide crystalline particles via a sintering method. This choice is clever, as sintering induces the development of crystalline defects that propagates over significant length scales. The subsequent 3D analysis of topological crystalline defects and their interactions highlights the power of the method and its capability to address crystalline defects analysis in a novel and elegant manner.

We thank the referee for their very positive feedback, we are glad to hear that they consider the results impactful and exciting.

Up to this point, a few different strategies were proposed to address the question of crystalline distortion characterization. On the one hand, Bragg coherent diffraction imaging based approaches [Ulvestadt et al., Science 2015, Li et al. Nature Commun 2021 and ref. 10 of the manuscript] collect the coherent diffraction signal measured in the vicinity of a Bragg peak to extract the crystalline displacement, from which lattice rotations and dislocations can be identified and characterized. They provide a highly sensitive and highly spatially resolved description of the crystalline structure, but within a limited crystalline distortion range (typical lattice mis-orientation below 2-3°). On the other hand, diffraction contrast tomography, proposed in the late 2000 [Ludwig et al., J. Appl. Cryst. 2008 and Ref 15 of the manuscript], is able to probe polycrystalline material presenting fully mis-oriented grains. However, the sensitivity of the method prevents the detailed analysis of the intra-grain distortion analysis. Another relevant approach, which the authors did not refer to, is dark-field microscopy [see e.g., Simons et al., Nature Commun. 2015], where an optical element is placed behind the sample, along a Bragg diffraction peak signal produced by one of the crystallites. The extensive scanning process grants access to a 3D map of the crystalline axis orientation, offering high sensitivity to crystalline distortion. However, the spatial resolution is constrained by the limitations of the X-ray optics.

We thank the referee for this overview of the existing techniques for the mapping of orientations and microstructure on the nanoscale. We have extended our introductory paragraph on lines 61-72 on page 2 to provide a more comprehensive overview of the available techniques:

“So far, **high spatial resolution microstructure mapping** has been possible with electron-based techniques such as Transmission Electron Microscopy^{7,8} (TEM) and Electron Back-Scatter Diffraction¹³ (EBSD), **achieving** sub-10 nm spatial resolution with planar measurements. However, as these measurements are limited to materials of thickness on the order of 100 nm, destructive sectioning methods are currently required to acquire full 3D orientation maps **of extended volumes**. The non-destructive imaging of **the crystallographic orientation of micrometre-thick materials has been addressed with diffraction contrast tomography [J. Oddershede et al. Integr. Mater. Manuf. 11 (2022)] and dark-field microscopy [H. Simons et al. Nat Commun. 6 6098 (2015)]**. However, these techniques are limited by the X-ray optics, with a typical spatial resolution of hundreds of nanometres, and so far have only been used to characterise highly crystalline samples. Crystal

orientation and strain mapping has also been demonstrated with Bragg Coherent Diffractive Imaging [A. Ulvestad et al. *Science* 348, 1344 (2015)] (BCDI) and Bragg Ptychography [S. O. Hruszkewycz et al. *Nature Mater* 16, 244 (2017) F. Mastropietro et al *Nature Mater* 16, 946 (2017), P. Li et al. *Nat Commun* 12 7059 (2021)] , where it is possible to obtain a precise mapping of the crystal orientation with a spatial resolution of tens of nanometres, albeit in an effectively single crystal object of low defect density [S. O. Hruszkewycz et al. *Nature Mater* 16, 244 (2017), A. Ulvestad et al. *Science* 348, 1344 (2015), F. Mastropietro et al *Nature Mater* 16, 946 (2017), P. Li et al. *Nat Commun* 12 7059 (2021)].”

With XL-DOT, the portfolio of crystalline microscopy is enriched by a method, which is more sensitive than diffraction contrast tomography, which overcomes the thickness limit of coherent diffraction imaging microscopy and which provides a better spatial resolution than the one imposed by the optical element of dark-field microscopy. Therefore, one can expect that new questions in material science at the nanoscale will be soon addressed, like the mechanism responsible for platelet stacking in columnar nacre.

We thank the reviewer for their positive assessment of the impact of XL-DOT in the broader context of the existing capabilities.

For these reasons, I fully support the publication of this article in *Nature*. However, several issues should be addressed in order to improve the understanding of the potential impact of the method and its domain of application, which I think the manuscript fails to convey. I detailed them below:

-1. - Detailed analysis of the limits of the method:

In its present form, a detailed analysis of the method limits is missing. Angular sensitivity and spatial resolution are discussed in details, and a series of tests and analysis are provided to support the conclusions. They were highly appreciated.

- Regarding spatial resolution, several analyses are included to discuss in details this question. This is totally fair and needed. I would appreciate that the spatial resolution discussion also includes an analysis on a smaller crystallite, similarly to what was done in Figure S8. This would allow evaluating whether the spatial resolution is a global parameter or whether it depends on the size of the scattering object.

We thank the referee for the suggestion to include further analysis of the spatial resolution. We have identified a suitable region, and measured edge profiles to assess the local spatial resolution. As for the previously presented example, the edge profiles exhibit sharp changes in the orientation with a 10-90% length of 47-65 nm. This is consistent with the calculated spatial resolution that was presented in the previous version of the manuscript and has been included in Supplementary Figure S8. Although, in general, spatial resolution is not a global parameter, it appears that it has a consistent value within our sample. The updated Figure S8 is also included in the next page of this document. The figure caption changes, affecting lines 996 to 998, are provided below the figure.

"Supplementary Figure S8: [...] the resolution of the XL-DOT reconstruction is given by the difference between the 10% Δ and 90% Δ distances. **The resolution was also calculated for a smaller grain, shown in e) orientation view and f) equivalent x-y azimuthal angle plot. g,h) Line profiles corresponding to the lines of the same colour in (f) demonstrating 47 nm to 65 nm resolution, similar to the resolution of the larger grains in (a-d).** Also provided are Fourier shell correlation (FSC) curves (i-k) of each orientation scalar component (x, y, z), respectively."

I-2. The energy dependence is not really discussed. In particular, how accurate should be the energy positioning with respect to the chosen edge? What happens if the material is composed of different compounds (or even polymorphs) with the same edge but different charge states / slightly varying energy profile at resonance? (for instance if the material is composed of aragonite and calcite sub-structures).

a) We thank the referee for highlighting this important point. The energy positioning is indeed important, as the linear dichroism is a resonant effect and therefore energy dependent. While energy positioning with respect to the chosen edge should be as accurate as possible to maximise the XLD signal, absorption edge features are frequently rather broad (the FWHM of the XLD peak for our sample is approximately 3 eV) [A. C. Stiffler et al. *Am. Chem. Soc.* **140** 11698 (2018), H. Ade and B. Hsiao *Science* **262** 1427(1993)]. Thus, there is a certain flexibility both in terms of energy position and required X-ray energy resolution to perform XL-DOT measurements. Notably, any deviation from the optimal measurement energy results in a decay of the XLD signal and a greater signal-to-noise ratio.

To address these remarks on energy positioning and resolution, we have included the following on lines 601-608 of the Methods section:

“Although XL-DOT measurements should ideally be performed at the X-ray energy of an absorption edge where the linear dichroic contrast is the strongest to maximise contrast in the projections, the extent, in terms of the x-ray energy, of the spectra at the absorption edge that display an associated dichroism can be quite large. For instance, the full width at half maximum of the near-edge peak in our V_2O_5 spectra used for XL-DOT is approximately 3 eV, meaning that even an X-ray energy resolution of up to 3 eV would be sufficient for XL-DOT measurements, albeit with a decreased signal-to-noise ratio. There is therefore a degree of flexibility in terms of the required energy position and energy resolution for XL-DOT measurements.”

We have also added a remark in the caption of the Supplementary Figure S4 on lines 947-949:

“The full width at half maximum of the peak is approximately 3 eV, indicating that a higher energy resolution is not required to perform XL-DOT on this material”.

b) The multicomponent or polymorph consideration is an excellent point to consider. While the XL-DOT electron density tomograms can readily be used for component and polymorph identification and segmentation (for example, the electron density of calcite is $0.82 \text{ n}_e\text{Å}^{-3}$, that of Aragonite $0.88 \text{ n}_e\text{Å}^{-3}$), the retrieval of the orientation field associated with the XLD contrast can be a challenge, albeit resolvable.

If the energies are close to each other, i.e. within a few eV, a single XL-DOT measurement is able to recover the orientation field of both polymorphs/components. If the energies are further apart, a secondary XL-DOT measurement at the energy exhibiting the maximum linear dichroic signal of the second polymorph would be required. To address this potential issue, we are planning to combine XL-DOT with sparse X-ray transmission near-edge spectroscopic tomography [Zirui Gao et al. *Sci. Adv.* **7** (2021), N. Ishiguro and Y. Takahashi *J. Appl. Cryst.* **55** 929 (2022)]. The resulting retrieval of XL-DOT datasets across an entire absorption edge, will allow us to retrieve the orientation field for each of the components/polymorphs present in a single measurement.

We address the XL-DOT sampling requirements for composite materials consisting of multiple materials or polymorphs in the newly added Supplementary Note 3:

“*Supplementary Note 3: XL-DOT composite material considerations and unit cell orientation: For the examination of composite materials, consisting of multiple, linear dichroism exhibiting materials and/or polymorphs, as for example encountered in biominerals or geological samples [A. C. Stiffler et al. *Am. Chem. Soc.* **140 11698 (2018), D. Zhou et al. *J. Phys. Chem. B* **112**, 13128 (2008)], it might not be possible to retrieve the orientation field of each material from a single XL-DOT measurement. In this case, multiple XL-DOT measurements might be needed to retrieve the orientation field of each material to highest accuracy. While the electron density tomograms can be used for material identification and segmentation, XL-DOT measurements at different X-ray energies are needed to retrieve the orientation field of each material if their associated spectral features, which exhibit linear dichroism, are distinctly separated. Should the features fall within a couple eV of each other, and depending on the instrument’s energy resolution, a single XL-DOT measurement is sufficient to recover the orientation field of the different materials simultaneously [G. Fevola et al. *Phys. Rev. Research* **2** 013378 (2020)].”***

I-3. Regarding the dichroic contrast induced by the crystalline orientation: I understand that delivering the reference spectra presented in Figure S4 is crucial. It allows providing the value of the factor A in equation S1 (it fixes the amplitude of the dichroic contrast for the two extreme orientations).

However, delivering these spectra requires accurate crystal orientation positioning and intensity normalization. How is it done in practice?

We thank the referee for highlighting this point. In fact, during the tomographic reconstruction, we do not constrain the magnitude of the orientation: this is left as a free parameter. As a result, it is not crucial to have a quantitative estimate of the contrast amplitude before performing the XL-DOT.

The spectrum is rather used to identify the optimal energy for which the dichroism is strongest, i.e. to determine the energy at which the XL-DOT measurement will be performed. For XL-DOT, these do not need to stem from a single crystal or specifically prepared sample but, instead, can be directly acquired from the to-be-measured sample by acquiring a series of XANES projections or images of the sample at the start of a beamtime [Z. Gao et al. Chem. Commun. **56** 13373 (2020)].

We have clarified this point by explicitly stating in the Methods section of the manuscript on lines 486-489:

$$f = f_0 + f_{lin}(\vec{E} \cdot \vec{a})^2 \quad (2)$$

“[equation (2) is included here for context] **During the reconstruction process, the magnitude of the XLD contrast, corresponding to f_{lin} , was not constrained and was therefore also optimised during gradient descent. As a result, it is not necessary to predetermine the f_{lin} value.**”

I-4 Furthermore, only samples that present an optical anisotropic behaviour in the x-ray regime can be investigated. How large is this class of materials?

All non-cubic crystalline, magnetic, ferroelectric and low-dimensional materials (for example, graphene, as well as selected coordination complexes and molecular arrangements/networks), can exhibit linear dichroism to some degree. [J. Stöhr “The Nature of X-Rays and Their Interactions with Matter” ch.7, S. Polisetty et al. J. Phys.: Condens. Matter **24** (2012), H. Ade and B. Hsiao Science **262** 1427(1993), C. Jansing et al. J. Phys.: Conf. Ser. **712** 012031 (2016)].

Of particular interest, and particularly relevant for future operando measurements, are a large fraction of transition metal oxides, sulphides, carbonates and phosphates, that represent a substantial fraction of heterogeneous catalysts, energy storage cathode and solid electrolyte materials, as well as biominerals and cementitious materials. Additionally, as the magnetic anisotropy is a source of linear dichroism, XL-DOT can be employed to map the Néel vector in antiferromagnets in 3D that, until now, has not been possible with existing techniques. As a result, the class of materials that can be targeted by XL-DOT is extensive. We have highlighted this in the manuscript on lines 299-301 as follows:

“Linear dichroism is exhibited by a large variety of materials, ranging from non-cubic crystalline materials to magnetic (**ferro-, ferri- and antiferromagnetic**), ferroelectric, **and low-dimensional materials, to selected non-crystalline alloys, coordination complexes and molecular arrangements / networks** [H. Ade and B. Hsiao Science **262** (1993), A. C. Stiffler et al. Am. Chem. Soc. **140** 11698 (2018), S. Polisetty et al. J. Phys.: Condens. Matter **24** (2012), C. Jansing et al. J. Phys.: Conf. Ser. **712** 012031 (2016), H. Jani et al. Nature **590** 74 (2021), N. Krins et al. ChemRxiv. 2024-2gjfk, J. Stöhr “The Nature of X-Rays and Their Interactions with Matter” ch.7].”

I-5. Temporal resolution : XL-DOT is expected to be compatible with operando approaches (see e.g., lines 57 and 286-292 of the manuscript). However, the acquisition procedure is rather time consuming. What was the total acquisition time of the presented result (and can you add the

information in the manuscript)? How can the acquisition time be reduced in the future and which reducing factor to expect? What temporal resolution could be targeted in fine?

We thank the referee for highlighting this point. The total acquisition time for our measurement was approximately 85 hours. However, the time taken purely for the measurements was ~24 hours. This discrepancy is largely due to missing automation in these first experiments. Although 85 hours might seem excessive when considering operando measurements, significant improvements can be expected in the future, as has been the case for, e.g., SAXS tensor tomography [M. Liebi et al. *Nature* 527, 349 (2015)], where a decrease from 35.5 hours to 1.2 hours was reported [arXiv:2406.13238 [physics.med-ph]] due to automation and optimisation in data acquisition and reconstruction. A similar reduction in acquisition time is expected for XL-DOT.

Opportunities to reduce the acquisition time, both immediate and long-term, are as follows:

a) The presented measurement is “oversampled”, and reconstructions of a reduced dataset indicate that similar results can be obtained with 50% of the acquired data. A detailed response including illustrations is provided under Query 2 of Referee 1 on Pages 3-5 of this reply.

b) Our numerical simulations, and published theoretical considerations, [M. A. Marcus *Opt. Express* 30 39445 (2022), Y. H. Lo et al. *P. Natl. A. Sci.* 118 (2021)] suggest that the multi-axis measurements can be replaced or compensated by measurements with additional linear x-ray polarisations (set, for example, using an Advanced Planar Polarised Light Emitter undulator). This would dramatically simplify data acquisition and the required instrumentation, and would increase the acquisition speed.

c) XL-DOT, in view of the often-weak X-ray linear dichroic contrast, is particularly photon hungry. As a result, very long scanning acquisition times of up to 240 seconds per projection are currently needed. The upgrade to 4th generation synchrotron light sources, taking as an example the cSAXS beamline in the SLS-2, is expected to provide a 40x increase in coherent flux. This directly leads to shorter exposure times, or increased resolution. Further advancements can be expected from positioning instrumentation [M. Odstrcil et al. *J. Synchrotron Rad.* 26 504 (2019)] and acquisition scheme developments, e.g. fly scan [X. Huang et al. *Sci Rep* 5, 9074 (2015)], and multi-beam ptychography [M. Lyubomirskiy et al. *Sci Rep* 12 6203 (2022), arXiv:2402.12082v2 [physics.app-ph]].

With these advances, we believe that an XL-DOT dataset will be measurable in less than 10 hours in the near future (~2025-2026).

Further advances in the imaging geometry, such as those implemented to carry out laminography [C. Donnelly et al. *Nat. Nano.* 15 356 (2020)], would remove the need for sample reorientation between tilt axes, allowing for significant time saved, ideal for the implementation of magnetic and electro-chemical operando measurements.

Finally, the acquired tomogram in an XL-DOT dataset exhibits heavily correlated signal changes—a set of grey level intensity variations in absence of sample deformations — that would benefit significantly from sparse tomographic acquisition and reconstruction, [Z. Gao et al. *Sci. Adv.* 7 (2021), Z. Gao Thesis ch6] likely saving about 80-90% of the required projections.

We have added the requested information on the measurement time and a brief outlook on how to increase the speed of measurements to the method section, lines 550-563:

“Current XL-DOT acquisition time and future prospects: The total acquisition time for the XL-DOT dataset used in the presented work was around 85 hours, including sample tilting, changing the

polarisation, and alignment and deadtime overheads. The pure measurement time, however, was only ~24 hours. This discrepancy is largely due to the lack of automation. There exist a number of opportunities to reduce the acquisition time, as follows:

1. Reducing oversampling: reconstructions using 50% of the tomograms provide similar results (Figure S14).

2. Automation and imaging geometry: the measurement of intermediate linear x-ray polarisation angles [M. A. Marcus (2022) and Y. H. Lo (2021), S. Sasaki Nucl. Instrum. Meth. A 347 83 (1999), V. Scagnoli et al. J. Synch Rad 16 (2009)] and/or use of the laminography geometry [Donnelly, C., et al. Nat. Nano. 15 (2020)], will eliminate the majority of the current acquisition overheads.

3. The increase in coherent flux expected from fourth generation synchrotron light sources promises to reduce scan times for radiation hard materials [M. Odstroil et al. J. Synchrotron Rad. 26 504 (2019)].

4. Further innovations such as multi-beam ptychography and sparse tomography offer routes to even faster data acquisition [M. Lyubomirskiy et al. Sci Rep 12 6203 (2022), Zirui Gao et al. Sci. Adv.7 (2021)], so providing acquisition times compatible with operando measurements. [K. Aliyah ACS Appl. Mater. Interfaces 16, 25938(2024), Donnelly, C., et al. Nat. Nano. 15 (2020)].”

II - Comparison to other methods:

In order to provide a fair assessment of XL-DOT with respect to the other crystalline microscopy methods, I believe a dedicated paragraph is needed in the discussion part. In its present version, this question is briefly addressed in a paragraph at the end of the manuscript (lines 280-285). This part should be improved, with clear references and positioning with respect to x-ray dark-field contrast microscopy, x-ray diffraction contrast tomography, Bragg CDI and Bragg ptychography. In particular, the strain seems to be not accessible with XL-DOT, is it correct?

We thank the referee for their suggestion to broaden the discussion of existing techniques. We have extended the introductory paragraph on lines 61 to 72:

“So far, **high spatial resolution microstructure mapping** has been possible with electron-based techniques such as Transmission Electron Microscopy^{7,8} (TEM) and Electron Back-Scatter Diffraction¹³ (EBSD), **achieving** sub-10 nm spatial resolution with planar measurements. However, as these measurements are limited to materials of thickness on the order of 100 nm, destructive sectioning methods are currently required to acquire full 3D orientation maps of **extended volumes**. The non-destructive imaging of **the crystallographic orientation of micrometre-thick materials has been addressed with diffraction contrast tomography** [J. Oddershede et al. Integr. Mater. Manuf. 11 (2022)] and **dark-field microscopy** [H. Simons et al. Nat Commun. 6 6098 (2015)]. However, these techniques are limited by the X-ray optics, with a typical spatial resolution of hundreds of nanometres, and so far have only been used to characterise highly crystalline samples. Crystal orientation and strain mapping has also been demonstrated with Bragg Coherent Diffractive Imaging [A. Ulvestad et al. Science 348, 1344 (2015)] (BCDI) and Bragg Ptychography [S. O. Hruszkewycz et al. Nature Mater 16, 244 (2017) F. Mastropietro et al Nature Mater 16, 946 (2017), P. Li et al. Nat Commun 12 7059 (2021)] , where it is possible to obtain a precise mapping of the crystal orientation with a spatial resolution of tens of nanometres, albeit in an effectively single crystal object of low defect density [S. O. Hruszkewycz et al. Nature Mater 16, 244 (2017), A. Ulvestad et al. Science 348, 1344 (2015), F. Mastropietro et al Nature Mater 16, 946 (2017), P. Li et al. Nat Commun 12 7059 (2021)].”

And updated the discussion paragraph, lines 287-297

“XL-DOT takes an important position amongst **current microstructural characterisation techniques, bridging the gap between the high spatial resolution imaging of nanoscale samples with BCDI,**

photoemission electron microscopy (PEEM) and electron microscopy techniques, and the lower spatial resolution imaging of larger samples accessible by diffraction contrast, dark-field tomography and tensor tomography^{10,40-42}. With the ability to image larger samples at high spatial resolution, XL-DOT offers the non-destructive examination of nanoscale defects within system-representative sample volumes^{7,10,12-15}. In contrast to the probing of long-range order with diffraction techniques^{13,8,10,27,15,24,41,42}, XL-DOT probes the short-range order of a material through linear dichroism^{16,32}. We note that, while the spectroscopic nature of XL-DOT presents additional opportunities, the material of interest must exhibit X-ray linear dichroism and comply with X-ray transmission requirements (see Methods).”

In particular, the strain seems to be not accessible with XL-DOT, is it correct?

Strain affects the local electronic and structural environment around the probed/absorbing atoms. As such, with linear dichroism, the strain component associated with the probed electronic transition is in principle accessible [Cao, W. et al. App. Surf. Sci. **265** 358 (2013)], albeit as a relatively weak contribution. In our manuscript, we have not addressed the presence of strain fields but, in theory, we would expect that indications of the strain field could be identified from the magnitude of the reconstructed orientation. Although beyond the scope of this current manuscript, it is certainly possible that, in the future, XL-DOT could be applied to map strain in three dimensions. Measurements would likely need to be combined with resonance or spectral tomography, (i) to obtain local electron density and chemical element concentration values as “strain normalisation factors” and (ii) as the width/, intensity and shape of the absorption edge convey further information about the crystallinity and strain of the probed material [Zirui Gao et al. Sci. Adv. **7** (2021)].

Since these are tentative ideas, we did not add a comment about the strain to the manuscript.

III - Other major remarks:

- Please provide the definition of the topological charge for non-specialist readers.

We thank the referee for the suggestion, which will improve the accessibility of the manuscript. We’ve added the following explanation on how the topological number which was calculated using the winding number around the topological defect in lines 574 to 579 in the methods section on page 21:

“Analysis of topological defects: The topological charge can be determined by considering the winding number associated with a given topological defect. The winding number corresponds to how the crystallographic orientation changes when moving around a circle enclosing the defect in a clockwise manner. For the comet (trefoil) defect, the c-axis rotates clockwise (anticlockwise) by +180° (-180°) for one complete revolution. As the crystallographic orientation has completed half a revolution of a full circle (360°), the topological numbers $\pm 1/2$ are assigned to them.”

- Line 294: what do you mean by orientation of amorphous material? By definition, an amorphous material does not have long range ordering.

We thank the referee for highlighting that our wording was confusing.

By amorphous, we were referring to the group of materials that can possess an order parameter (long-range ordering over a well-defined distance) in the absence of crystal ordering. Linear dichroism, for instance, can result due to magnetic ordering (X-ray magnetic linear dichroism), even in systems where a crystal structure is absent, such as alloys [J. Kuneš et al. J. Magn. Magn. Mater. **272** 2146 (2004)], which we referred to in the previous version of the manuscript as amorphous (we now refer to these

as non-crystalline). Additionally, linear dichroism is present in polymeric coordination complexes and molecular arrangements/networks which can exhibit linear dichroism [H. Ade and B. Hsiao *Science* **262** (1993)]. We have updated the text of the manuscript to refer to non-crystalline materials (lines 39, 77, 294 in the previous version of the manuscript), which is a large class of materials that can be imaged using XL-DOT.

- Line 309: How did the FIB damage the sample? Can you briefly describe?

During the fabrication of the sample, FIB milling was used to define the cylindrical shape. A typical side effect of using a Ga ion FIB processing is the implantation of Ga ions in the surface region. From the reconstructed electron density analysis, it was observed that FIB damage was only present on the outer layers of the sample with a thickness of around 90 nm. In these regions, the damage could be identified in the electron density reconstruction due to the increased electron density, which is again consistent with gallium implantation, as well as material redeposition and sample amorphization. Although the outermost layers were not included in the analysis of the XL-DOT, Supplementary Figure S9 was updated to demonstrate the effect of the FIB processing, which we also include here for the convenience of the referee, with the relevant changes to the caption:

“Supplementary Figure S9: [...] (c) Slice through the electron density tomogram, with regions of polymer (i) and the defect-rich V₂O₅ (ii) marked accordingly. The increased electron density present on the outermost layers of the pillar (~90 nm-thick) was attributed mainly to gallium implantation (but also amorphization and material redeposition), which resulted from the FIB milling during sample preparation. As gallium has a higher electron density than V₂O₅ (see Table S1), the electron density of the affected area is greater than in the central parts of the sample, which remained unaffected.”

This surface damage is a common observation in Ga-FIB sample preparation [Zirui Gao et al. *Sci. Adv.* 7 (2021) - supporting information].

- Density: how was the electron density of the defect-rich V₂O₅ established in Table S1? In Figure S9, it seems there is a shift between the mark ‘(ii)’ corresponding to the established density and the maximum of the electron density distribution. Is it significant?

We thank the reviewer for the suggestion to clarify this point. The electron density (ρ) was calculated using the following physical parameters:

$$\rho = \frac{\text{Density}}{\text{Molar Density}} N_A * (\text{Sum of Atomic Numbers in the Formula Unit})$$

Knowing that oxygen vacancies are common in V_2O_5 [D. O. Scanlon et al. J. Phys. Chem. C **112**, 9903 (2008)], the formula can be adjusted to account for them as follows: (assuming the volume of the unit cell does not change)

$$\rho = \frac{3.357 \text{ g/cm}^3}{181.862 \text{ g/mol}} 6.022 \times 10^{23} (23 * 2 + (5 - x) * 8),$$

Where x is the average number of oxygen vacancies per unit cell. When $x = 0$ (no vacancies), the equation evaluates to 0.96 \AA^{-3} (pristine V_2O_5 , given in Table S1). Following the PXCT measurement, the electron density of the measured sample was instead 0.85 \AA^{-3} , which corresponds to, on average, 1.2 oxygen vacancies per unit cell.

Our observations of 1.2 oxygen vacancies per unit cell align well with experimental reports of an average of 1.14 oxygen vacancies per unit cell observed at 400°C [Q.-H. Wu et al. Appl. Surf. Sci. **236** 473 (2004)]. Given the high temperature sintering step during sample fabrication (590°C), favouring the formation of additional oxygen-vacancy defects, the deviation from the estimated value is not significant.

We have made the following changes to the supplementary information on lines 885-895 to provide additional context for the reader:

“Supplementary Note 2: Electron density estimation of oxygen-vacancy rich V_2O_5 : The electron density, ρ , was determined using tabulated density and molar density values according to the equation

$$\rho = \frac{\text{Density}}{\text{Molar Density}} N_A * (\text{Sum of Atomic Numbers in the Formula Unit})$$

Additionally, for oxygen-vacancy rich V_2O_5 the following equation was used

$$\rho = \frac{3.357 \text{ g/cm}^3}{181.862 \text{ g/mol}} 6.022 \times 10^{23} (23 * 2 + (5 - x) * 8)$$

where N_A is the Avogadro constant and x is the average number of oxygen vacancies per unit cell. From PXCT measurements, we determine the average number of oxygen vacancies per unit cell to be 1.2. Given the additional annealing step triggering oxygen vacancy formation, the observed 1.2 oxygen vacancies per unit cell agrees with experimental reports of 1.14 oxygen vacancies per unit cell [Q.-H. Wu et al. Appl. Surf. Sci. **236** 473 (2004)]. For the pre-experimental estimate of the electron density of oxygen-vacancy rich V_2O_5 , (Table S1) a 0.85 oxygen vacancies per unit cell were assumed based on a reference sample.

IV Additional minor remarks

- Typos on line 505 and 508, on the density units
- The exponent has been re-written with the correct font.

Referee #2 (Remarks on code availability):

I did not review the code in details

- the readme file indicates to run the reconstruction.m, which I did not find
- 3D tomographic reconstructions are provided, as indicated in the readme routine.

If I have a bit more information from the authors, I would be happy to review the code in more details

We thank the referee for highlighting this issue and checking our reconstruction code.

Referee #3 (Remarks to the Author):

Apseros et al implement linear dichroic ptychographic tomography with very high resolution in an impressive experimental tour de force. The work is technically impressive and interesting but for the reasons listed below, I am not convinced about the general applicability, uniqueness and impact of the method. Therefore, I cannot support publication of this paper in Nature.

We thank the referee for their time and effort spent on reviewing the manuscript and bringing up important questions. In addition to our reply to each of the Referee's points presented after each paragraph below, we first address the general applicability and uniqueness of the technique in comparison to the presently available microstructural characterisation techniques.

In terms of applicability, it turns out that a wide range of materials exhibit linear dichroism, and that the applicability of XL-DOT is not as constrained as the Referee currently believes. While XL-DOT is not currently suitable for the examination of isotropic materials, the majority, if not all non-isotropic crystalline materials, as well as magnetic, ferroelectric and low-dimensional (e.g. graphene) materials, and coordination complexes and molecular arrangements will exhibit linear dichroism to some degree [J. Stöhr "The Nature of X-Rays and Their Interactions with Matter" ch.7, H. Ade and B. Hsiao. Science **262** (1993)]. Of particular societal relevance are transition metal oxides, sulphides, carbonates and phosphates, as they represent a substantial fraction of heterogenous catalysts, energy storage cathode and solid electrolyte materials, biominerals and cementitious materials. Additionally, as magnetic anisotropy is a source of linear dichroism, XL-DOT can be employed to map the Néel vector in antiferromagnets in 3D, which until now has not been possible. With this multitude of examples that are highly relevant to some of the most important research areas for the development of a sustainable society, our assertion that XL-DOT will be of high impact as a characterisation technique is justified.

In terms of uniqueness, XL-DOT is exceptional because of its combined capabilities, neatly fitting a niche and complementing existing microstructural characterisation methods. In particular, XL-DOT provides a quantitative, 3D, non-destructive, inter- and intra- granular compositional and microstructural characterisation of spatially extended, polycrystalline or composite samples with nanometre resolution. By implementing XL-DOT with ptychography, the spatial resolution is not limited by the scanning step size or X-ray optics, so that high resolution maps can be obtained in a time-efficient manner. Furthermore, XL-DOT easily extended to include chemical characterisation through spectral tomography.

Of course, the technique, as rightly pointed out by the three Referees, comes with certain application requirements/restrictions: the sample must exhibit linear dichroic contrast that requires the use of specific X-ray energies, and this gives an associated limit on the sample thickness that can be measured in transmission. However, all characterisation methods come with limitations.

We hope that we have now satisfactorily addressed these limitations and avenues for future improvements in the revised manuscript.

1. The authors state that no other methods except TEM give 3D orientation information on the nanoscale. However, dark field X-ray microscopy (Simons, H., King, A., Ludwig, W. et al. Dark-field X-ray microscopy for multiscale structural characterization. *Nat Commun* 6, 6098 (2015)) and Bragg ptychography (Mastropietro, F., Godard, P., Burghammer, M. et al. Revealing crystalline domains in a mollusc shell single-crystalline prism. *Nature Mater* 16, 946–952 (2017)) give such information at similar resolution as established in the present manuscript and additionally directly provide information on e.g. strain. Neither method is discussed by the authors, which is a significant problem.

We thank the referee for pointing out that the introductory line in the paragraph may have been misinterpreted to read that only electron-based techniques can be used to characterise the microstructure: this was not the intended meaning of our sentence, as we also described the use of diffraction contrast tomography:

[In the previous version of the manuscript]: “So far, mapping of the local crystal orientation has been possible with electron-based techniques such as Transmission Electron Microscopy (TEM) and Electron Back-Scatter Diffraction (EBSD), which can achieve sub-10 nm spatial resolution with planar measurements. ... The non-destructive imaging of micrometre-thick materials has been addressed with Diffraction Contrast Tomography, but this technique is limited to the size of the X-ray probe, ...”

We have reformulated lines 61-63 to emphasise that electron-based techniques provide high spatial and angular resolution of the orientation:

“So far, **high spatial resolution microstructure mapping** has been possible with electron-based techniques such as Transmission Electron Microscopy (TEM) and Electron Back-Scatter Diffraction (EBSD), **achieving** sub-10 nm spatial resolution with planar measurements.”

In the previous version of the manuscript, we mentioned diffraction contrast tomography as an available x-ray technique for imaging extended samples. We appreciate the referee’s suggestion to include a complete overview of the available techniques, and thus have expanded the introductory discussion to also include dark-field microscopy, Bragg Coherent Diffractive Imaging and Bragg Ptychography on lines 65 to 72 on page 2 as follows:

“The non-destructive imaging of **the crystallographic orientation of micrometre-thick materials has been addressed with diffraction contrast tomography [J. Oddershede et al. *Integr. Mater. Manuf.* 11 (2022)] and dark-field microscopy [H. Simons et al. *Nat Commun.* 6 6098 (2015)]. However, these techniques are limited by the X-ray optics, with a typical spatial resolution of hundreds of nanometres, and so far have only been used to characterise highly crystalline samples. Crystal orientation and strain mapping has also been demonstrated with Bragg Coherent Diffractive Imaging [A. Ulvestad et al. *Science* 348, 1344 (2015)] (BCDI) and Bragg Ptychography [S. O. Hruszkewycz et al. *Nature Mater* 16, 244 (2017) F. Mastropietro et al *Nature Mater* 16, 946 (2017), P. Li et al. *Nat Commun* 12 7059 (2021)] , where it is possible to obtain a precise mapping of the crystal orientation with a spatial resolution of tens of nanometres, albeit in an effectively single crystal object of low defect density [S. O. Hruszkewycz et al. *Nature Mater* 16, 244 (2017), A. Ulvestad et al. *Science* 348, 1344 (2015), F. Mastropietro et al *Nature Mater* 16, 946 (2017), P. Li et al. *Nat Commun* 12 7059 (2021)].”**

2. The authors indicate that the proposed method is widely applicable to map orientation. However, orientation is obtained by proxy in the proposed approach: the authors map the X-ray linear dichroism, which is related to orientation but not in a manner that – in general – can be simply linked to the signal. This is explored extensively in the PEEM work of PUPA Gilbert who demonstrated the need to measure control samples to link orientation and dichroism. In the present example, the authors make use of the special symmetry and structure in V2O5 that defines the relationship

between orientation and dichroism. However, such a relationship does not exist in general. This limits the scope of the proposed method and is not even mentioned in the manuscript by the authors.

XL-DOT maps the orientation field of the probed anisotropy by measuring changes in the linear dichroism intensity. J. Stöhr in the “The Nature of X-Rays and Their Interactions with Matter” in ch.7, describes the linear dichroic contrast as the “search light effect”: the linear polarisation acts as a search light for the resonant bond that it is parallel to. Absorption is strongest when the resonant bond is parallel to the polarisation and weakest when perpendicular. This relationship between the dichroism and the responsible anisotropy or orientation in principle applies to all cases of natural linear dichroism and the origin of the linear dichroic contrast can be inferred from the unit cell structure in crystalline materials and from the structural motif in non-crystalline materials.

This is explored extensively in the PEEM work of PUPA Gilbert who demonstrated the need to measure control samples to link orientation and dichroism.

In the experiment where the *c*-axis of apatite in bone was mapped [Stifler, A. C. et al. Am. Chem. Soc. 140 (2018)], it appears that the calcium can have one of two possible coordination numbers, or “environments”, with each exhibiting different linear dichroic spectra. In this case, in order for the authors to disentangle the two sources of linear dichroic signal corresponding to these two environments, which are in close proximity in energy, spectral pre-characterisation of reference samples was necessary.

In V₂O₅ and other materials, where only a single environment is present, the separation of multiple dichroic signals is not necessary. In our manuscript (lines 347-363 in latest manuscript), the relationship between the crystal orientation and the linear dichroism is supported by the cited density functional theory (DFT) work as well as spectral measurements of V₂O₅ performed by Horrocks et al. [J. Phys. Chem. C. 120 (2016)]. In this case, the K-edge pre-edge peak energy that we use for the measurement corresponds to the *c*-axis orientation of V₂O₅. This is also the case for the study of Apatite by A. C. Stifler et al. [Am. Chem. Soc. 140 (2018)]; where they “...acquire a PIC map [...] assuming a Malus-law dependence of intensity on the angle between the polarisation and the *c*-axis.” demonstrating that, despite the complex spectral shape of the linear dichroism, the concept of the “search light effect” still holds.

In summary, we agree that it is important to understand the relationship between the linear dichroism and the crystal orientation in a sample, and that this can be complex and energy dependent. However, we do not agree that “such a relationship does not exist in general” or that it “limits the scope of the proposed technique”. This relationship is already well-known for many materials and crystal structures from X-ray spectroscopy and, as demonstrated by the cited DFT work we use here, can be well described by theory.

We amended a section of methods to make the connection between the measured “bond-anisotropy” orientation to crystallographic orientation clearer, lines 365 to 372:

“Origin of Linear Dichroism in α -V₂O₅: [...]

In the above-described relationship between the polarisation state of the illumination and the probed asymmetry or anisotropy, the linearly polarised light acts as a “search light” for the resonant bond that it is parallel to. This relationship applies, in principle, to all cases of natural linear dichroism [J. Stöhr “The Nature of X-Rays and Their Interactions with Matter” in ch.7, A. Hsiao and B. Hsiao Science 262 (1993)]. The connection between the probed anisotropy orientation and the unit cell orientation of the material, can be obtained through the use of reference samples, as showcased in 2D linear dichroic microscopy applications [A. C. Stifler et al. Am. Chem. Soc. 140

(2018)] and is already available in the literature for numerous materials. It can also readily be determined with prior knowledge of the material's crystal structure (or molecular arrangement) [Horrocks et al. *J. Phys. Chem. C.* **120**].”

3. The claim on page 2 that the method can be used to map amorphous materials in 3D does not seem warranted.

Since we are using transmission-based techniques for XL-DOT, in this case ptychography, it is possible to image all components of the pillar, such as the polymer, which is a non-crystalline material. As the polymer can be measured using transmission, it is possible to extend this to polymers exhibiting linear dichroism: indeed, X-ray linear dichroism of polymer samples has been demonstrated in [A. Hsiao and B. Hsiao *Science* **262** (1993)]. We note that, by amorphous materials, we referred to materials that do not possess crystalline order but can have other ordering that can be measured, such as magnetic ordering, which can also be probed using linear dichroism [J. Kuneš et al. *J. Magn. Magn. Mater.* **272** 2146 (2004)]. As we have realized that this definition was not quite correct, we have reworded the manuscript to refer to “non-crystalline” materials instead of amorphous materials exhibiting linear dichroism (lines 39, 77, 294 in the previous version of the manuscript).

4. The method, as I understand it, requires the measurement of multiple ptychography datasets (8!). The current measurement involved a dose of a GGy. In practice, this means that there will certainly be beam damage in many sample classes, e.g. polymers, which is already a limiting factor in many ‘simple’ ptychography experiments. Again, this is an issue that is not really discussed but that in practice will limit the methods applicability. This certainly applies to the rather broad sweeping concluding remarks on page 13: “This opens the door... bone”. The authors would have to demonstrate some of these capabilities or remove these claims.

We appreciate the concerns of the referee when it comes to the number of datasets required as well as the dose needed to perform XL-DOT.

a) As this was a first demonstration, we made sure to obtain more than enough data for a successful reconstruction. To address the number of tilts necessary for a reconstruction, we have now performed numerical simulations as well as an XL-DOT reconstruction with only two tilts (four tomograms). We find that similar results, albeit lower quality in terms of spatial and angular resolution, can be obtained. A detailed response including illustrations is provided under Query 2 of Referee 1 on pages 3 - 5 of this reply.

b) We also note, that thanks to constant technical development, the acquisition of several ptychographic tomograms within a single beamtime, or the equivalent number of projections when considering larger samples or time-resolved experiments, is becoming increasingly possible. [S. Shirani et al. *Nat. Commun.* **14** 2652 (2023), Z. Gao et al. *Sci. Adv.* **7** (2021)]

c) Dose efficiency is of paramount importance when working with polymers or biological samples which can rapidly degrade when exposed to x-rays. For our investigation, in addition to the oversampling to ensure a successful proof of principle demonstration, we did not prioritize dose efficiency as V_2O_5 does not typically degrade when exposed to X-ray radiation with the doses implemented. When it comes to measuring samples that are prone to radiation damage, adjustments (preventative measures) and compromises (spatial resolution and sample size) can be made. This consideration applies to all X-ray nanotomography techniques and there are a number of available options to reduce the deposited dose, and limit or account for potential radiation damage, during

acquisition and post-processing, that are applicable to XL-DOT utilizing ptychography as the imaging modality:

- A switch from in-focus nanobeam scanning probes to micrometre-sized scanning probes, as well as a relaxation in the scanning pattern density (number of scanning points), which leads to a reduction in peak dose and cumulative dose. This change appears to be particularly effective for the measurement of organic-rich or polymeric samples [E. F. Garman Acta. Crystallogr. D. **66** 339 (2010)].
- The measurement of tomograms under cryogenic conditions and in an inert atmosphere has been shown to be effective in limiting radiation damage [M. Holler et al. Rev. Sci. Instrum. **89**, 043706 (2018), S. H. Shahmoradian et al. Sci Rep **7**, 6291 (2017)].
- Borrowing from electron microscopy, one can deposit conductive metallic thin films (1-5 nm thick) on the surface of non-conductive samples and radiation-sensitive samples such as metal-organic frameworks. In this way, charge accumulation and temperature hot spots can be avoided.
- In some cases, radiation damage is not preventable as pointed out by the reviewer. In such cases, a way forward could be through the use of recently-developed advanced reconstruction algorithms which can account for and follow such damage, for example via the implementation of nonrigid X-ray nanotomography [M. Odstrcil et al. Nat Commun **10**, 2600 (2019)].

To better reflect the importance of the deposited X-ray dose, and the available measures to prevent, limit and account for radiation damage, we have added the following information to the Method section of the manuscript on page 19, lines 511 - 517:

“Dose Estimation: The total deposited radiation dose over the duration of the experiment and the entire volume of the V_2O_5 pillar is approximately 10^9 Gy. This estimate is based on the sample’s mass density and the average flux density per projection⁸¹. **No actions were taken to limit the dose, as V_2O_5 is not known to degrade under the current experimental conditions [Z. Gao et al. Sci. Adv. **7** (2021), Z. Gao et al. Chem. Commun. **56**, 13373 (2020)]. For radiation-sensitive materials, preventative measures can be employed to mitigate or account for potential radiation damage [M. Odstrcil et al. Nat Commun **10**, 2600 (2019)]. Dose-limiting options include scanning and projection sparse acquisition schemes, [Z. Gao et al. Sci. Adv. **7** (2021), O. Townsend et al. Opt. Express **30**, 43237 (2022)], reducing the total deposited dose, changes to the ptychography acquisition such as out-of-focus acquisition with micrometre-sized scanning probes, reducing both total and peak dose per area, as well as the implementation of cryogenic and inert atmosphere measurement conditions [M. Holler et al. Rev. Sci. Instrum. **89** 043706 (2018), E. F. Garman Acta. Crystallogr. D. **66** 339 (2010)].”**

This certainly applies to the rather broad sweeping concluding remarks on page 13: “This opens the door... bone”. The authors would have to demonstrate some of these capabilities or remove these claims.”

We respectfully disagree with the Referee since all requirements needed to perform such measurements on bone have been previously demonstrated.

Apatite in human bone and mouse enamel, as well as in parrotfish bone, dentin, enameloid and collagen were shown to exhibit measurable X-ray linear dichroism [N. Krins et al. ChemRxiv. 2024-2gjf, A.C. Stifler et al. Am. Chem. Soc. **140** (2018)]. Stifler et al. [Am. Chem. Soc. **140** (2018)] used X-

PEEM to measure c-axis orientation of apatite in bone in 2D, confirming the linear dichroic signal required for XL-DOT.

Biominerals, including bone and synthetic inorganic-organic of similar organic content, have previously been measured using ptychographic tomography by us and others, both under atmospheric and cryo-conditions with roughly similar X-ray doses [M. Liebi et al. *Nature* 527, 349 (2015)].

In order to clarify this point, the statement in the manuscript was changed to be more explicit and additional citations were added, lines 305-306:

“This would [...] characterisation of biominerals [Stifler, A. C. et al. *Am. Chem. Soc.* 140 (2018), N. Krins et al. *ChemRxiv*. 2024-2gjfk] [...], both the organic and inorganic parts, and [...]”

5. On page 15: “Although XL-DOT...” does not mention the PEEM work of PUPA Gilbert (which is cited later), which certainly merits mention here – it is 2D but only requires a simple polished surface.

Although we appreciate the work of Pupa Gilbert, we cannot directly fulfil the referee’s request at this manuscript section. The isolated mention of PEEM would undermine the pioneering work of Ade [H. Ade and B. Hsiao *Science* 262 1427(1993)] regarding X-ray linear dichroic microscopy (cited in the abstract). In the original manuscript, several referrals to the work Gilbert are given alongside the works of Ade and Collins in Supplementary Note 1. We believe that it is more appropriate to cite these leading works on X-ray linear dichroic microscopy together.

To place these efforts more prominently in the main text we moved this section to the Referee suggested location which now reads as follows, lines 389 to 394:

“The angular dependence of the linear dichroism has previously been used in a microscopy context, in particular in X-ray linear dichroism microscopy with secondary imaging modalities such as photoemission electron microscopy, to provide a 2D spatially-resolved microstructural characterisation tool^{16,32,33,40,66,67}. The reader is directed to the initial work of Ade¹⁶ and the more recent works of Gilbert^{40,68–70} and Collins^{66,67,71}. In the current work, we have developed the capability to map the orientation in 3D by combining X-ray linear dichroism microscopy with PXCT (XL-DOT).”

6. On page 18, the authors mention dose, but the actual time the measurements took is never mentioned. Given the claims that the method should be applicable in situ/operando, the real measurement time is important and should be given.

Thank you for pointing out this oversight. We apologise for the omission, and this information has been added, as well as a discussion concerning opportunities to speed up the measurements. A detailed response is provided under Query I-5 of Reviewer 2, pages 13 - 14 in this document.

7. In the section “Measurement Error Estimation” on page 18, the authors report the average measure phase shift to be 5.8E-6 with a standard deviation of 1.3E-4 rad. This would correspond to a standard deviation that is 1.5 orders of magnitude larger than the signal – please explain?

In this part of the manuscript, the aim was to calculate the error in the linear dichroic contrast. To do this, we reconstructed the tomograms obtained using linear horizontal (LH) and linear vertical (LV) polarisations individually. After obtaining the dichroic component $XLD = LH - LV$ by subtraction a region of air surrounding the sample was selected within the XLD tomogram. We expect that this region of air should have an average dichroism of 0, and therefore we have computed the average value of XLD contrast from the value present in the selected area. Assuming normally distributed data, the standard

deviation of the contrast would be an estimate of the error. In the manuscript, the mean was included for a complete picture and to demonstrate the value is indeed close to 0 but was ultimately not relevant to the discussion. More relevant to the discussion are the average values of the electronic scattering ($1.5\text{E-}2$ rad) and the linear dichroism inside the sample ($1.8\text{E-}3$ rad), which are both at least one order of magnitude greater than the standard deviation. To avoid confusion, we have removed the mean value from the manuscript and only kept the standard deviation as a measure of error in the linear dichroic contrast. These changes affected lines 538 to 544 on page 20.

“To estimate the uncertainty in the detected LD, i.e., spatial variations in the pre-edge peak intensity, we **independently** reconstructed the **LV and LH phase tomograms** with the sample at a fixed sample tilt, and then subtracted them from each other. We then isolated a region of air and calculated the standard deviation in the phase shift associated with the voxels in this region. **This standard deviation of the phase associated with the air region corresponds to the uncertainty of the dichroic signal.** Based on this procedure, **the uncertainty of the dichroic signal is found to be 1.3×10^{-4} rad**, which corresponds to a refractive index decrement, δ , **error of 1.9×10^{-7} .**”

8. Taken together, the manuscript presents a technical tour de force but does in my opinion not demonstrate a widely applicable advance or novel insights into an important material. The method has rather significant limitations that the authors downplay or ignore, and relevant competing methods are not discussed. I therefore cannot support publication of this paper.

We thank the referee for their feedback, and we appreciate the time and effort that they have put into this thorough review of the manuscript. We believe the quality and content of the manuscript has significantly improved as a result.

Regarding the specific criticisms of the referee:

1. **Does not demonstrate a widely applicable advance:** In our reply, we have shown that the class of materials exhibiting linear dichroism is extensive. The technique exploits resonant edges, which uniquely provides element specific information, and can also be combined with spectral tomography for the chemical characterisation of the sample.
2. **Novel insight:** We have demonstrated the high spatial resolution as well as the advanced capabilities of XL-DOT to identify structural and topological defects in V_2O_5 . V_2O_5 serves as a crucial catalyst in the production of sulfuric acid, an industry worth \$10 billion. Our aim is to extend the application of this technique to the examination of pristine and aged (in-use) catalysts with industrial collaborators to elucidate failure and optimization of these catalyst. These measurements would be performed *ex-situ* in the near future and *operando* in the intermediate future. Our previous industry collaborations using similar technique developments have led to practical changes in both catalyst design and industrial use protocols and conditions.
3. **Relevant competing methods are not discussed:** We have included a comprehensive overview of the presently available techniques that can be used for microstructural characterisation, as well as the position XL-DOT takes with respect to them in the introduction and outlook.
4. **The method has rather significant limitations:** We have now added more explicit statements on the limitations of the XL-DOT technique to present the technique in a more balanced manner. However, we disagree that these limitations are significant. In fact, XL-DOT represents a versatile technique that can be applied to a wide variety of material systems.

We hope that, with these changes to the manuscript and our responses to the queries of the referee, the referee now considers our manuscript suitable for publication in Nature.

Author Rebuttals to First Revision:

We thank the referees and editorial staff for their positive reviews and feedback. The manuscript was revised in line with the suggestions from Referee 3 and in alignment with the editorial guidelines.

Provided below is a point-by-point response to the referees' comments and editorial requests. The comments are presented in black, the responses in blue. Changes to the manuscript are quoted below the respective replies. Changes to the manuscript text are highlighted in blue (first) and orange (second revision).

Referee #1

(Remarks to the Author):

The authors have sufficiently addressed all of my comments in their reply and in the changes made to the manuscript. The effort to add new figures and context greatly improves this impressive work.

We thank the referee for their constructive review and are pleased to hear that our revisions in the figures and text have addressed all of their comments.

Referee #2

(Remarks to the Author):

I am fully satisfied with the revised version proposed by Andreas Apseros et al. My questions have been considered with great care. The detailed answers are fully convincing. The modifications made to the manuscript fully take into account my remarks. The manuscript is clearly improved. It is now more balanced regarding its positioning with respect to the scientific framework and other methodologies and more impactful regarding the class of (crystalline and non-crystalline) materials the proposed method is able to address. I think it can be accepted for publication and I strongly support its acceptance in Nature journal.

We thank the referee for their constructive review and are glad to hear that our revisions have satisfied their concerns.

(Remarks on code availability):

I have run the code and encountered problem with GPU, as my machine is not equipped with GPU. However, the other parts of the code are working and are well documented.

Referee #3

(Remarks to the Author):

The revised manuscript of Apseros et al. is significantly improved compared to the first version of the manuscript. I really appreciate how the authors now address limitations of XL-DOT and inclusion of other methods. This has resulted in a much more balanced manuscript. I am therefore now prepared to possibly support acceptance of the manuscript in Nature provided suitable responses to a few additional concerns are provided:

We thank the referee for these comments and glad to hear that our revisions have improved the manuscript. We address their remaining concerns in this reply.

1. The authors have in their response compared reconstructions with a varying number of tilts (new Figure S14). This is most welcome and an excellent addition to the good existing analysis of reconstruction fidelity. However, I don't follow their conclusion that the two-tilt reconstruction "there appears to be an overall agreement in terms of the reconstructed orientation with the four-tilt

reconstruction." While it is certainly true that a proportion of the orientation map coincides, there are important differences, some of which I have encircled in the attached file. If one only had access to the 2-tilt reconstruction, these would be interpreted as smaller grains or similar, possibly resulting in erroneous conclusion on the properties of the sample. My reading of figure S14 is rather that it demonstrates that at least 3 tilts are needed for appropriate reconstruction of orientation. This does not detract from XL-DOT as a technique. But it leads to a question: do the authors foresee a possibility to also extract a local confidence map, where one could evaluate the 'noise' in the reconstruction? This would be a most interesting addition, but I fully understand if it is beyond the scope of the present work.

2. However, I do think that the presentation of Figure S14 should be changed – in my reading it does not demonstrate that two tilts are enough, rather that four are much better. The authors should either do a full (3D) quantitative analysis of the two vs four tilt reconstruction or simply conclude that at present four tilts are needed but that further work on acquisition schemes may reduce the number of required tilts. I really think that the authors (and XL-DOT) are better served by being a bit conservative here.

We thank the referee for their recommendation. We have updated the Figure caption and discussion in line with the referee's suggestion. Specifically, we mention explicitly that a two-tilt reconstruction can present artefacts, compared to four tilts, and further mention that other acquisition schemes that can reduce setup complexity and acquisition time are being explored.

Caption: "From the plot it is evident that four tilts or more yield optimal results, while a preliminary reconstruction can be obtained from as few as two tilts. ...Although the reconstruction with only two tilt axes has an increased angular uncertainty, lower spatial resolution, and is prone to artefacts, there appears to be an overall agreement in terms of the reconstructed orientation with the four-tilt reconstruction."

Discussion: "Nevertheless, we note that a preliminary reconstruction can be obtained with as few as two tilt axes (Figure S14). Moreover, we envisage that the use of alternative geometries ..."

3. In line 285-286, I suggest adding also Bragg ptychography so that the phrase becomes "...of nanoscale samples with BCDI, Bragg ptychography, photoemission electron microscopy (PEEM)..." for balance in presentation of the techniques.

We have adapted the text as suggested.

4. Line 302-305: the authors retain their statement that XL-DOT will be suitable for orientation mapping of biominerals, now writing "...both the organic and inorganic parts...". The cited example in reference 40 deals with PEEM studies of Ca dichroism while reference 46 deals with optical LD of polymers, as I read it (As I understand reference 46, wavelengths 170-320 nm). For the organic component, the cited preprint, reference 46, studies optical LD. In my reading, the authors state that XL-DOT can be applied in 3D on samples of e.g. bone using the optical dichroism – or do they envision hard X-ray LD from HCNO-containing polymers to be strong enough to make the experiment successful? Maybe I am simply not understanding what the authors intend to say, but please provide an estimate of the transmission possible with a realistic biomineral sample to test feasibility of LD measurements of the polymers in several micron sized biomineral specimens (bone as many biominerals is white due to very significant light scattering that limits transmission in the visual domain and proteins absorb strongly around 280 nm). What sample sizes do the authors envision will be measurable?

Biomaterial experiments would be performed using tender (Ca K-edge at 4038 eV and P K-edge at 2145 eV) and soft X-rays (e.g., various L-edges and notably the C K-edge at 284 eV). The isolated citation of the optical wavelength demonstration was indeed misleading; we have replaced the reference with citations [H. Ade, B. Hsiao *Science* **262** 1427 (1993)] and [R. S. K. Lam *et al. ACS Chem Biol* **7**(3) 476 (2012)], which report on LD at the carbon edge, the latter in collagen molecules. Additionally, an example of LD in bone at the Ca K and L-edges was added to the citation [C. A. Stifler *et al. J. Am. Chem. Soc.* **140** 47 (2018)].

While selected biominerals can naturally possess dimensions compatible with XL-DOT measurements in the tender and soft X-ray range, e.g. magnetosomes and coccolith, others including bone need "thinning" to be compatible. This is common in biomineral X-ray microscopy and for X-ray

measurements in general [N. K. Witting *et al.* Tomography of Materials and Structures **5** 100027 (2024), <https://doi.org/10.1016/j.tmater.2024.100027>]. We calculate an estimate of the thicknesses that can be targeted as follows:

In the case of bone, assuming 65 wt.% inorganic content, primarily hydroxyapatite ($\sim\text{Ca}_{10}(\text{PO}_4)_6(\text{OH})_2$), 35 wt.% organic content, mainly type I collagen ($\sim\text{C}_7\text{H}_{11}\text{N}_1\text{O}_4$), an average density of $1.2 \text{ g}\cdot\text{cm}^{-3}$ (between cortical and trabecular), and a 10-20% transmission as cut-off criterion, samples with a circular cross-section up to:

- ~40 μm in diameter could be investigated at the Ca K-edge, probing the inorganic part.
- ~15 μm in diameter could be investigated at the P K-edge, probing the inorganic part.
- ~1 μm in diameter could be investigated at the C K-edge, probing the organic part.

Estimates were obtained using [<https://henke.lbl.gov/>], with the transmission taken at the absorption peak, giving conservative estimates given pre-edge and phase contrast microscopy measurement options, as recently reported by J. Neethirajan *et al.* in Phys. Rev. X **14**, 031028 (2024), <https://doi.org/10.1103/PhysRevX.14.031028>.

We have updated the sentence as follows:

This would, for example, allow, with nanoscale precision, the 3D characterisation of (sectioned) biominerals [37,42], both the organic and inorganic parts, and synthetic polymers¹⁶ in solar cells, as well as secondary phases located in grain boundaries or material inclusions within grains. For the biominerals and analogous systems, as investigations would involve tender and soft X-rays, e.g. Ca and C K-edges, sample diameters are likely to be restricted to sub 5 μm in diameter.

6. Do the authors have an estimate of what 'concentration' the target element should have to allow XL-DOT measurements? In many biocomposites, the concentration of the target elements, e.g. Ca, are significantly lower than in the present example of V₂O₅.

We thank the referee for highlighting this important point. Determining a general threshold for the concentration of a material is challenging, as it is dependent on multiple parameters (flux, imaging setup, material composition and density, anisotropy, to name a few). Nevertheless, we have calculated some estimates that are specific to our measurements in the following:

The critical concentration for element detection can be estimated to correspond to a dichroic magnitude (difference between tomograms taken with different polarizations) of at least twice the reconstruction error. The dichroic contrast of the V₂O₅ is 1.8×10^{-3} rad and the noise in the reconstruction is an order of magnitude weaker at 1.3×10^{-4} rad. As a result, in V₂O₅ our dichroic contrast is 12 times the error. One can estimate that, if all other parameters are held constant, the concentration of V can be decreased by a factor of 6 and still be measurable. This limit corresponds to a density of 2 V atoms per nm^3 .

We note that common biomineral materials such as CaCO₃ (calcite ~ 11 Ca atoms nm^{-3}), SiO₂ (quartz, ~ 17 Si atoms nm^{-3}) or Ca₅(PO₄)₃(OH) (hydroxyapatite, ~ 19 Ca atoms nm^{-3}) exhibit equivalent concentrations significantly above this threshold.

We further note that these estimates are measurement specific, and assume a fixed acquisition time, flux, etc. As mentioned in our manuscript, with the increase in flux at 4th generation synchrotrons, we can expect the sensitivity of our measurements to increase, and for weaker dichroic signals to become accessible. We have added estimates of the threshold concentration to the error estimation section in the methods as follows:

The critical concentration of elements in order for them to be detected approximately corresponds to a dichroic magnitude (difference between tomograms taken with different polarizations) of at least twice the reconstruction error. The dichroic contrast of V_2O_5 is 1.8×10^{-3} rad and the noise in the reconstruction is an order of magnitude weaker at 1.3×10^{-4} rad. As a result, in V_2O_5 , our dichroic contrast is 12 times the error. One can estimate that, if all other parameters are held constant, the concentration of V can be decreased by a factor of 6 and still be measurable.

7. Concerning the discussion (lines 309-320) on in operando: it is in principle true that almost any experimental technique can study sufficiently slow dynamics. The relevant question becomes whether the time scales that can be studied are relevant for the stated materials classes. Even given the foreseen improvements in technology, XL-DOT will – as I read the response from the authors – remain a relatively slow (hour-scale) technique. This does not make it bad, but it does place limits on the applicability for in operando studies. I urge the authors to either acknowledge this (stating the ‘slow’ can be followed or that there will a time resolution limit) or (in my view the better option) leave out the section on in operando measurements. The information obtained in XL-DOT is interesting enough on its own without this perspective outlook on in operando studies.

The referee is correct that, for the foreseeable future temporal measurements on the hour timescale, or quasi-static process investigations, will be the focus. Investigations will include the exploration of microstructure changes as a function of external conditions, such as controlled or intermittent pressure, temperature, state-of-charge, atmosphere, hydration level or magnetization changes. As recommended, we include this current constraint into the text of the manuscript:

"Here, one can envisage following the evolution of the microstructure during material synthesis, under annealing conditions, albeit limited in temporal resolution by the XL-DOT acquisition time. By performing quasi-static operando measurements, the exploration of microstructural changes under external stimuli including the application of pressure, and changes in the atmosphere or temperature, will become possible, opening the door to, for example, the study of solid state batteries, fuel cells or heterogeneous catalysts under operational conditions."